# Direct impact of COVID-19 by estimating disability-adjusted life years at national level in France in 2020

Romana Haneef[1]*, Myriam Fayad[2], Anne Fouillet[2], Cécile Sommen[2], Christophe Bonaldi[2], Grant M. A. Wyper[3], Sara Monteiro Pires[4], Brecht Devleesschauwer[5,6], Antoine Rachas[7], Panayotis Constantinou[7], Daniel Levy-Bruhl[8], Nathalie Beltzer[1], Anne Gallay[1]

1 Department of Non-Communicable Diseases and Injuries, Santé Publique France, Saint-Maurice, France, 2 Department of Data science, Santé Publique France, Saint-Maurice, France, 3 School of Health & Wellbeing, University of Glasgow, Glasgow, United Kingdom, 4 National Food Institute, Technical University of Denmark, Lyngby, Denmark, 5 Department of Epidemiology and Public Health, Sciensano, Brussels, Belgium, 6 Department of Translational Physiology, Infectiology and Public Health, Ghent University, Merelbeke, Belgium, 7 Department of Strategy, Studies and Statistics, French National Health Insurance: Caisse nationale de l'assurance maladie (Cnam), Paris, France, 8 Department of Infectious Diseases, Santé Publique France, Saint-Maurice, France

* Romana.HANEEF@santepubliquefrance.fr

**Data Availability Statement:** All relevant data are within the manuscript and its Supporting Information files.

## Abstract

### Background

The World Health Organization declared a pandemic of coronavirus disease 2019 (COVID-19), caused by severe acute respiratory syndrome coronavirus 2 (SARS-CoV-2), on March 11, 2020. The standardized approach of disability-adjusted life years (DALYs) allows for quantifying the combined impact of morbidity and mortality of diseases and injuries. The main objective of this study was to estimate the direct impact of COVID-19 in France in 2020, using DALYs to combine the population health impact of infection fatalities, acute symptomatic infections and their post-acute consequences, in 28 days (baseline) up to 140 days, following the initial infection.

### Methods

National mortality, COVID-19 screening, and hospital admission data were used to calculate DALYs based on the European Burden of Disease Network consensus disease model. Scenario analyses were performed by varying the number of symptomatic cases and duration of symptoms up to a maximum of 140 days, defining COVID-19 deaths using the underlying, and associated, cause of death.

### Results

In 2020, the estimated DALYs due to COVID-19 in France were 990 710 (1472 per 100 000), with 99% of burden due to mortality (982 531 years of life lost, YLL) and 1% due to morbidity (8179 years lived with disability, YLD), following the initial infection. The contribution of YLD reached 375%, assuming the duration of 140 days of post-acute consequences

**Funding:** The author(s) received no specific funding for this work.

**Competing interests:** I have read the journal's policy and the authors of this manuscript have the following competing interests: Romana Haneef is the first and corresponding author of this paper, and is the section editor of "health information system" of "Archives of Public Health". Brecht Devleesschauwer is one of the co-authors of this paper, and is academic editor at PLOS ONE and the guest editor of the article collection on "burden of disease" of "Archives of Public Health". All other authors declare that they have no competing interests related to the work. This does not alter our adherence to PLOS ONE policies on sharing data and materials.

of COVID-19. Post-acute consequences contributed to 49% of the total morbidity burden. The contribution of YLD due to acute symptomatic infections among people younger than 70 years was higher (67%) than among people aged 70 years and above (33%). YLL among people aged 70 years and above, contributed to 74% of the total YLL.

## Conclusions

COVID-19 had a substantial impact on population health in France in 2020. The majority of population health loss was due to mortality. Men had higher population health loss due to COVID-19 than women. Post-acute consequences of COVID-19 had a large contribution to the YLD component of the disease burden, even when we assume the shortest duration of 28 days, long COVID burden is large. Further research is recommended to assess the impact of health inequalities associated with these estimates.

## Introduction

The World Health Organization declared a global pandemic of coronavirus disease 2019 (COVID-19), caused by severe acute respiratory syndrome coronavirus 2 (SARS-CoV-2), on March 11, 2020 [1]. In France, between January 10 and January 24, 2020 (the confirmation period of the first cases in France), nine possible cases were identified of which three cases were confirmed with COVID-19 [2]. Between February 2020 and the end of December 2020, there were two waves of the COVID-19 epidemic in France: the first in March to mid-May 2020; and the second wave in October to November 2020 [3]. To control the epidemic and keep the transmission rate low, the French government implemented two national lockdowns from March 17 to May 10, 2020 [4], and from October 28 to December 1, 2020 [5]. The French population was encouraged to maintain protective public health measures such as social/physical distancing, extensive testing and tracing followed by isolation of cases and quarantine of close contacts, wearing facemasks in public, and COVID-19 vaccination.

Monitoring population health is essential to identify where health improvements are required to inform targeted prevention and intervention strategies, and to guide policies for addressing unmet health needs and to reduce health inequalities. The disability adjusted life year (DALY) is one the most common summary metrics of population health, and has become a key metric for quantifying the burden of disease (BoD) [6, 7]. The standardized approach of the DALY allows for quantifying the combined impact of morbidity and mortality of diseases and injuries [7]. For morbidity, severity distributions and disability weights (DW) allow for modulation based on how debilitating living with disease is, whereas age-conditional life expectancies weight mortality, reflecting that deaths at younger ages have a higher public health impact [8]. The impact of the COVID-19 pandemic on health occurs through two main pathways: directly, as an infectious disease, and indirectly, as a risk factor, for example through an increase in mental health issues due to national lockdown or delays to surgery, follow-ups and diagnoses through restrictions to vital health care services [9]. Henceforth, it is essential to understand the extent of the direct impact of COVID-19 on population health. Thus far, the health impact of the COVID-19 pandemic measured in DALYs has primarily focused on the direct burden of COVID-19. To our knowledge, no study has measured the burden of disease due to COVID-19 in France. Therefore, this study aimed to estimate the direct impact of COVID-19 at the national level in France for the 2020 calendar year, to combine the

population health impact of infection fatalities, acute symptomatic infections and their post-acute consequences.

## Materials and methods

We adopted the European Burden of Disease Network consensus disease model for COVID-19 using an incidence-based approach to estimate DALYs due to COVID-19 [10]. DALYs measure the healthy life years lost due to diseases and are calculated by combining years lived with disability (YLD) and years of life lost to premature mortality (YLL). Estimates of BoD indicators were calculated at the French national level (including French metropolitan and French overseas regions). Data inputs on confirmed cases and deaths (both confirmed and suspected), covered the whole period of 2020. The first two suspected deaths related to COVID-19 (coded using International Classification of Diseases (ICD-10) code U07.2 as the underlying cause of death), were recorded on January 1, 2020.

### Morbidity data and years lived with disability

**SI-DEP (Système d'Information de DEpistage Populationnel).**    Since May 2020, the French public health agency, Santé Publique France (SPF), which has led COVID-19 surveillance activities in France, has access to a database recording biological test results of COVID-19 screening, including polymerase chain reaction (PCR) and antigen tests performed in all biological, medical city laboratories and pharmacies in France [11]. This database, records the number of new positive test cases with symptoms (symptomatic) and without symptoms (asymptomatic) and demographic data can be summarized by sex and age-group, ranging from 0 to 90 years and over. SI-DEP is used to monitor the incidence rate, positivity rate and screening rate based on the number of tested patients and the number of positive cases. These indicators are calculated at national and subnational levels and can be aggregated by age or sex for a specific time frame.

For this study, we included all cases that tested positive (with PCR or antigen tests). We applied a grace-period of 60 days, before considering additional positive tests from the same individual. The case definition of positive cases was based on the SI-DEP database as the 60 days period was considered in the calculation of the number of positive cases to take into account COVID-19 reinfections. This implies that if a person has a second positive test at least 60 days after a first positive test, then it will be considered as a new incident case and two incident cases would be included. If a person has two positive tests recorded in a period of fewer than 60 days, then they would only be counted once as an incident case [11]. There was missing information regarding the absence or presence of symptoms experienced (asymptomatic or symptomatic) for 14% of the positive cases in the SI-DEP database, which we imputed using proportional redistribution by age and sex.

**SI-VIC (Système d'Information pour le suivi des VICtimes d'attentats et de situation sanitaires exceptionnelles).**    This information system was created in 2016 following the late-2015 series of terrorist attacks in Paris. SI-VIC can be used in situations such as health emergencies, to facilitate support to patients and to monitor hospital care, such as the number of available hospital beds [12]. SPF has access to the data related to COVID-19 patients hospitalized with and without intensive care via the SI-VIC database, including their length of stay, since March 2020.

### Health states

According to the French data sources for COVID-19, different health states defined using the consensus model were considered [13] (Table 1). These health states represent the different

**Table 1. Health states, durations and disability weights for estimating the burden of COVD-19 in France [13].**

| Health state | Description | Duration | Disability weight (95% uncertainty interval) | |
|---|---|---|---|---|
| | | | | Source |
| Asymptomatic | Has infection but no symptoms | 7 days | 0 | |
| **Acute symptomatic COVID-19 infections** | | | | |
| Mild/Moderate | No smell No taste, diarrhea, sore throat, sneezing, coughing, fever, and, pneumonia, | 10 days | 0.051 (0.032–0.074) | (Salomon et al. 2015) |
| Severe | Hospitalised, non-intensive care | 12 days (mean) | 0.133 (0.088–0.190) | (Salomon et al. 2015) |
| Critical | Hospitalised with intensive care | 20 days (mean) | 0.655 (0.579–0.727) | (Haagsma et al. 2015) |
| **Post-acute consequences of COVID-19/Long COVID** | | | | |
| Post-acute consequences | Person still feels symptoms such as fatigue, insomnia, depression | 28 days | 0.219 (0.148–0.308) | (Salomon et al. 2015) |

severity levels considered for COVID-19 cases. For mild/moderate health states, we assumed that all the cases were with moderate symptoms and have the same disability weight. This data input from this health state obtained from SI-DEP. The data inputs for severe (hospitalized cases) and critical (hospitalized with intensive care) health states were obtained from the SI-VIC database.

**Duration.** The duration corresponds to the time from the onset of symptoms to the improvement of the patient's condition [14]. According to the French data available from different sources, the duration for each health state was defined as follows (Table 1):

*Asymptomatic and symptomatic (mild/moderate) cases*: Based on expertise from infectious disease specialists and clinical practitioners who were dealing with COVID-19 patients, a duration of 7 days was considered for asymptomatic cases. This was in accordance with the official duration of isolation of cases in France in 2020 [15]. We defined a duration of 10 days for mild/moderate cases and used the reference of the Dutch study [16].

*Severe and critical cases*: The duration for severe cases corresponds to the time between the first admission to conventional hospitalization and subsequent discharge from SI-VIC (death or return home) for patients admitted in 2020. The duration for critical cases corresponds to the time between the first admission in critical care and discharge from SI-VIC (death or return home) for patients admitted in 2020. We assumed that these patients (i.e., severe and critical cases) were discharged upon the improvement of their acute infections. We calculated the mean duration of severe and critical health states for 10 age groups (S1 Table).

*Post-acute consequences of COVID-19/Long COVID*: The post-acute consequences of COVID-19 are also called as long COVID. To estimate the post-acute consequences for COVID-19, we assumed that 1-in-7 (14.3%) acute symptomatic patients with mild/moderate, severe, and critical manifestations, would have suffered from post-acute consequences of COVID-19 for four weeks (28 days), based on previously calculated transition probabilities [17]. At the time this study was conducted, studies to collect data on post-acute consequences of COVID-19 in France are ongoing.

**Disability weights.** DWs reflect how debilitating each health state is estimated to be and are measured from 0 (no impact, perfect health) to 1 (equivalent to death). For different health states considered in this study, DWs were based on the Global Burden of Disease (GBD) 2019 DWs for infectious diseases [18, 19]. For critical care, the DW was based on the European Disability Weights Study [18, 19] (Table 1).

For each sex and age category, YLD were calculated by summing across all health states $i$, the product of the number of incident cases ($N_i$), duration ($D_i$) and the disability weight

$(DW_i)$, [13]:

$$YLD = \sum_i N_i D_i DW_i$$

Total YLD was obtained by summing age and sex specific YLD. We also described the YLD estimates among males and females according to mild/moderate, severe and critical cases.

## Mortality data and years of life lost

Mortality data were extracted from death certificates filled out by medical practitioners and was obtained from the French Epidemiology Center on medical causes of death *(Centre d'épidémiologie sur les causes médicales de décès*: *CépiDc)* [20]. This database is managed by the National Institute of Health and Medical Research and produces annual national statistics of medical causes of deaths in France, in collaboration with the National Institute of Statistics and Economic Studies (INSEE). The 'cause of death' section of the death certificate consists of free-text fields where physicians write down one or more causes of death. Then, CépiDc codes the medical causes of death (CoD) and identifies an underlying cause of death following the World Health Organization (WHO) rules, using the tenth revision of the ICD-10. The underlying causes of death are selected using an automatic system (IRIS software [21]) for coding and prioritizing multiple cause of death. It can take between four to six months for the free-text medical causes of death to be made available to epidemiologists, and even longer for subsequent coding. For 2020, the available ICD-10 codes were provisional, as coding was not fully completed at the time of our study. Our analysis was based on approximately 80% of deaths certificates coded with COVID-19 as the underlying cause, and 20% of death certificate that had a mention of COVID-19 (not-coded and included the free text). The search algorithm used to extract COVID-19 related terms from the free-text section of the death certificate, has a high performance.

COVID-19 deaths were defined as deaths reported with COVID-19 as the main underlying cause of death according to WHO recommendations. COVID-19 deaths were coded as having an ICD-10 code from U07.1 (virus identified) and U07.2 (virus not identified) as underlying or associated cause of death, based on guidance from World Health Organization [22]. In death certificates with a "confirmed COVID-19" cause (code U07.1), we could not ascertain if the confirmation was based on a PCR or antigen test. In our main analysis, we included deaths with COVID-19 as the main underlying cause of death.

For each age group and sex, YLL were estimated by multiplying the number of COVID-19 deaths $D$ by the remaining life expectancy $RLE$ from the GBD reference life table 2019 [23] that assigns the same value to both males and females.

$$YLL = D \times RLE$$

Total YLL due to COVID-19 was obtained by summing all sex and age specific YLL estimates.

We estimated absolute DALYs and DALYs per 100 000 inhabitants using French population estimates from INSEE as denominator, by age and sex in 2020. For the interpretation of results, we used an arbitrary choice of two age categories, $< 70$ and $\geq 70$ years. We also calculated the YLL per death by dividing the total YLL by the total deaths.

The study was based on aggregated and anonymous data from national mandatory databases (SIVIC: article L3131-9-1 and R3131-10-1 code de la santé publique; SIDEP: article 11 loi n˚2020–546 and article 8 and following décret n˚2020–551). In accordance with French law, SPF has been granted specific access to these national databases in order to carry out its

mission of health monitoring and health crisis response (article L. 1413–7 code de la santé publique). Therefore, no ethics approval or consent of participant was needed, as the processing made by SPF is provided for by law.

## Scenario analyses

We performed a sensitivity analysis by varying input parameters of the number of mild/moderate cases and the duration of symptoms, and by quantifying and comparing the uncertainty intervals of estimated YLD.

## Detection rates of undiagnosed/untested symptomatic cases

At the time of submission of this study, the true incidence of COVID-19 for the whole year of 2020 in France was not available. We hypothesized that a large proportion of the population with mild and moderate symptoms potentially remained undiagnosed/untested. Therefore, we increased the mild and moderate symptomatic cases by 50% and 75% (based on experts inputs), to assess the effect on YLD and DALYs estimates.

## Duration of mild/moderate cases

The duration of symptoms of a mild/moderate case was not recorded in SI-DEP. The duration estimates might be impacted by biases in recorded cases, i.e., shorter durations might not be picked up, so there is uncertainty in this parameter. To investigate the impact of the duration of symptoms on the final estimates of burden, we applied two different durations (i.e., 7 days and 14 days) for mild/moderate cases and measured the difference in YLD and DALY estimates.

**Post-acute consequences of COVID-19/long COVID.** The duration of post-acute consequences of COVID-19 was based on the transition probability calculated in a previous study [17]. At the time this study was performed, no data on duration of long COVID were available in France. Therefore, we assumed two scenarios, one with a threefold (i.e., 84 days) and another one with a fivefold (i.e., 140 days) increase in duration of post-acute consequences. The choice of 84 days and 140 days was based on published literature [24, 25] and experts' inputs.

**Using both underlying and associated causes of death.** We calculated YLL using COVID-19 deaths including both underlying and associated causes of death.

## Results

### Mortality and morbidity due to COVID-19

In 2020, 1 050 551 asymptomatic cases, 1 420 879 symptomatic cases, 138 303 severe cases with hospitalization and 25 850 critical cases with hospitalization and intensive care, were reported. In total, 72 735 deaths due to COVID-19 were reported as an underlying cause of death in 2020 in France. Of deaths, 51% were among men and 49% among women, with 39% recorded among the 80–89 age group.

### Years of life lost to premature mortality

In France in 2020, 982 531 YLL due to COVID-19 (1460 per 100 000), of which 422 747 years of life were lost by women (43%) and 559 784 years of life lost by men (57%). YLL among people aged 70 years and above, contributed to 74% of the total YLL. The people aged between 80–89 years had more than one-third (34%) of the total YLL (Fig 1) and women accounted a higher share of YLL than men (37% vs 31%) in the same age group. On the other hand, men

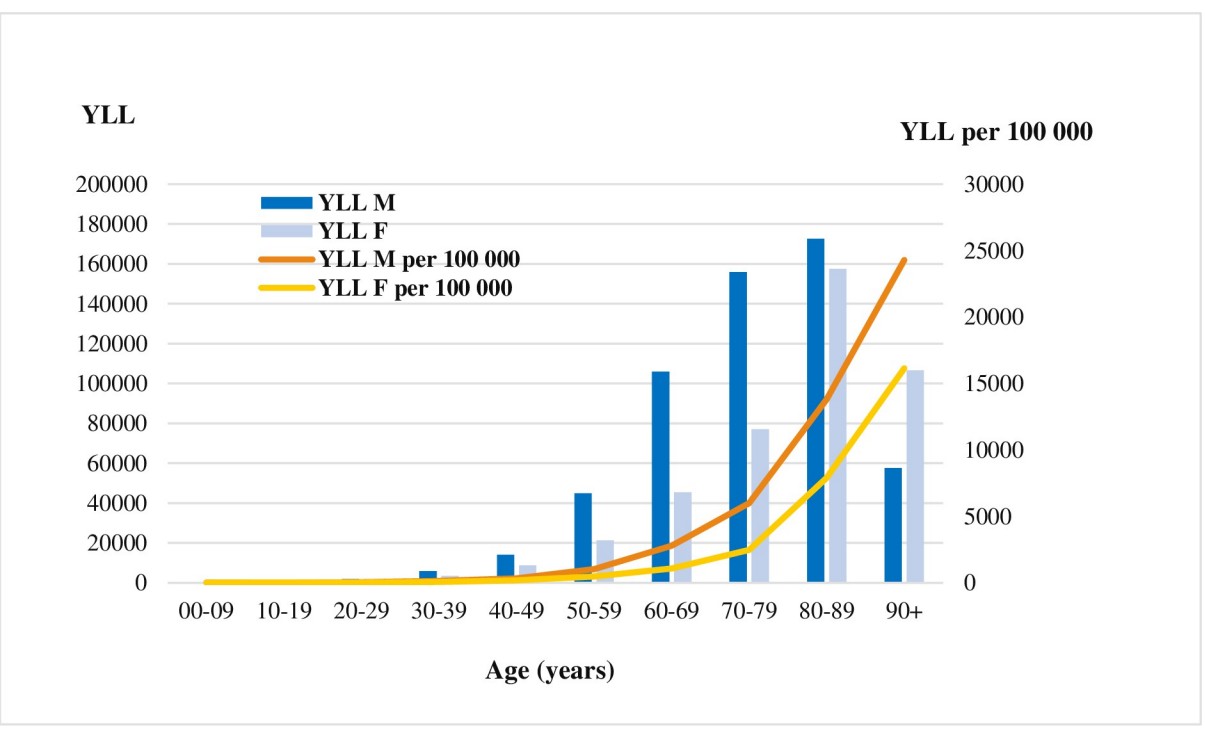

**Fig 1. Years of life lost due to COVID-19 in France in 2020 by age and sex.**

younger than 70 years accounted for a much higher share of YLL (31% of 559 784) than women of the same age (19% of 422 747). We estimated over 14 YLL for each death with COVID-19 in France in 2020 (15 YLL in women vs 19 YLL in men).

## Years lived with disability

A total of 4208 YLD (13 per 100 000) were estimated due to acute symptomatic COVID-19 infections including mild/moderate (2174 YLD), severe (835 YLD) and critical (1199 YLD) cases (Fig 2). Men between 50–79 years of age suffered higher YLD than women of the same age (1189 vs 783), particularly men between 70–79 years of age who were the sub-group with the highest YLD (i.e., 464). The percentage of YLD among people younger than 70 years was higher (67%) than those aged 70 years and above (33%). Contrary to YLL, among people younger than 70 years, women accounted for a slightly higher share of YLD than men (69% vs 66%).

The severity distribution of YLD varied among men and women (Fig 3). Overall, the proportion of YLD among women due to mild/moderate cases was 60% and 44% among men. Women younger than 70 years of age had a larger proportion of YLD due to mild/moderate cases than women aged 70 years and above (89% vs 11%). The proportion of YLD due to severe cases was similar in both sexes (20% women vs 19% men). YLD due to critical cases was higher among men (37%) than women (20%). A detailed description of YLD due to acute symptomatic COVID-19 infections in France in 2020 by age, sex and severity level is reported in S3 Table.

Finally, we estimated that approximately 226 660 people (14.3% of acute symptomatic infections, calculated based on published transition probabilities) suffered from post-acute consequences of COVID-19 or long COVID, which generated 3971 YLD. In total, 8179 YLD were estimated due to COVID-19, comprising of YLD due to acute symptomatic infections (4208; 51%) and YLD due to post-acute consequences of COVID-19 or long COVID (3971; 49%).

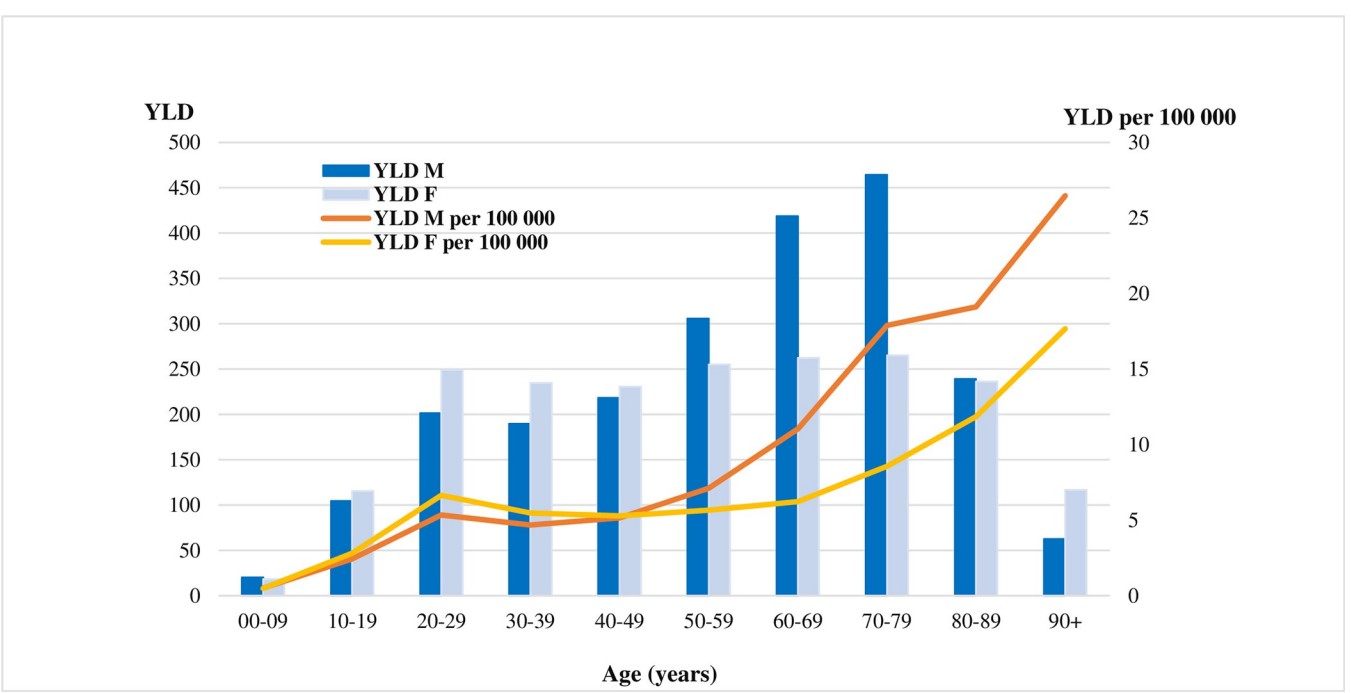

**Fig 2. Years lived with disability due to acute symptomatic COVID-19 infections in France in 2020 by age and sex.**

The proportion of YLD due to long COVID (3971 YLD) was higher among women than men (54% vs 46%).

## Disability-adjusted life years

In 2020, we estimated that there were 990 710 (1472 per 100 000) DALYs due to COVID-19. We observed that 99% of DALYs were due to mortality (982 531 YLL) and 1% was due to morbidity (8179 YLD). The percentage of DALYs due to acute symptomatic infections (i.e., including only YLD of mild/moderate, severe and critical cases [4208 YLD] and YLL estimates) were higher among men than women (57% vs 43%). People aged 70 years and above had a higher DALY estimates compared to people aged under 70 years (728 688 vs 258 051) (Fig 4).

## Scenario analyses

Fig 5 shows the impact of the analyses performed on estimated YLD and DALY (Table 2). *First*, increasing the number of positive cases by 50% and 75% led to an increase of 26% (5295) and 39% (5838) of YLD, respectively, as compared to the main results of YLD (4208). This increase led to an estimated 987 827 and 988 370 DALYs, respectively, as compared to the main results of DALYs (986 740). *Second*, varying the duration from 7 days to 14 days led to an increase of 17% in estimated YLD, from 3484 to 4933. *Third*, varying the duration for post-acute consequences of COVID-19 or long COVID cases to 84 days and 140 days, it led to an increase of 188% with a total of 11 417 YLDs and 375% with a total of 18 863 YLD, respectively. The duration of 140 days had the highest effect on DALYs (i.e., 1 001 394). A detailed analysis by sex is reported in S2 Table.

When using COVID-19 mortality estimates based on both underlying and associated causes of death, YLL estimates increased by almost 60%, compared to using only COVID-19 as the underlying cause of death (1 041 951 vs 982 531, respectively).

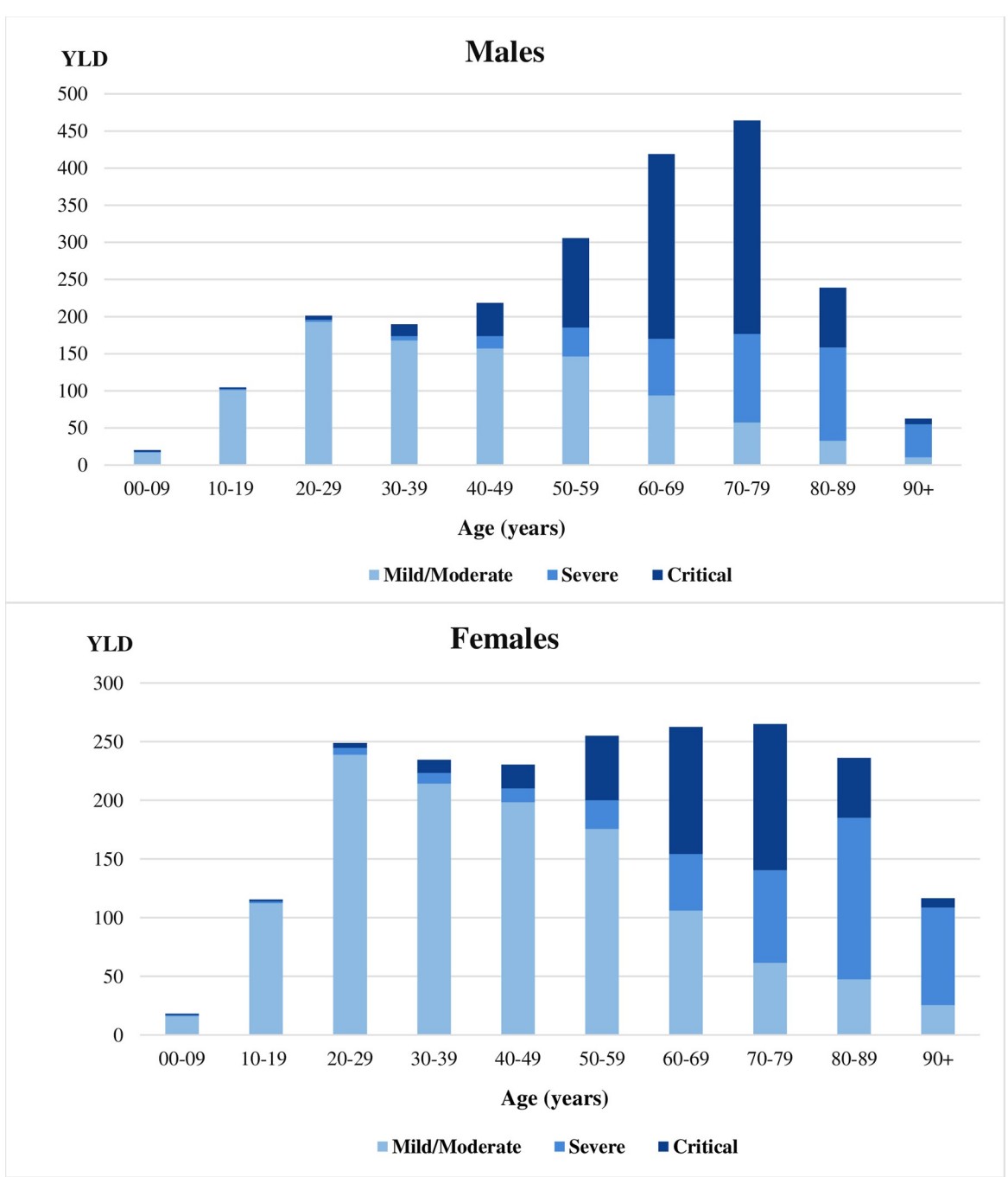

**Fig 3. Years lived with disability due to acute symptomatic COVID-19 infections in France in 2020 by age, sex and severity level.**

## Discussion

Our study is the first to estimate DALYs associated with the direct health impact of COVID-19 in France in 2020, the first year of the pandemic. The majority of population health loss was due to mortality, which contributed to 99% to the estimated DALY. This finding is in the context of long COVID being capped at a maximum of 140 days, but would still hold if long COVID duration was uncapped for the entire year. People aged 70 years and above had higher

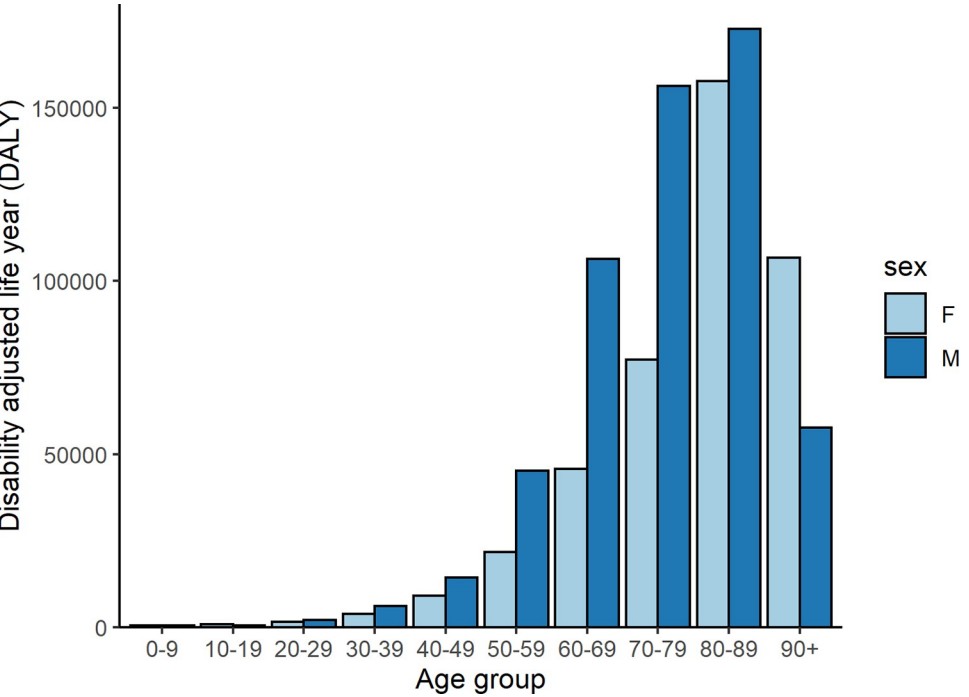

**Fig 4. Disability adjusted life years due to acute symptomatic COVID-19 infections in France in 2020 by age and sex.**

health loss due to mortality when compared to people aged younger than 70 years. On the contrary, people aged younger than 70 years had higher disability due to acute COVID-19 infections than those aged 70 years and above. Our analysis highlighted that even if long COVID resolves after less than six months, the post-acute consequences of COVID-19 had a large contribution to the YLD component of disease burden. Moreover, we observed that women had higher YLD due to the post-acute consequences of COVID-19 than men (2147 vs 1824). Other studies have also highlighted that females were more likely to have post-COVID syndrome than males [26–28]. However, population health loss due to mortality and morbidity due to COVID-19 infections, was higher among men than women.

In Europe, at the time this study was performed, DALY estimates have been published using the European Burden of Disease consensus disease model for COVID-19 in Germany, Scotland, Malta, the Netherlands, the Republic of Ireland and Denmark for 2020 [16, 29–33] (S4 Table). Outside Europe, Australia have also published DALYs estimates using the Burden of Disease consensus disease model for COVID -19 [34].

These countries made different choices such as the use of GBD reference life table compared to national life table; COVID-19 mortality definitions; durations used for each health state; and the consideration of post-acute consequences of COVID-19. For example, the Australia study applied the same assumption for the duration of hospitalized and critical cases (i.e., average length of stay) as France, but yielded different results. The average length of hospital stay among people aged 80 years and over was longer in France than in Australia (i.e., 20 days vs 12 days). These differences suggest that patients were critically ill and required longer time in intensive care in France than in Australia or they may have died sooner in Australia in critical care. The heterogeneity in health care provisions in hospitals, and health insurance system may also affect the length of stay in the hospital, and consequently YLD estimates. Overall, the length of stay in hospital increased with age in both countries. In France, YLL was higher

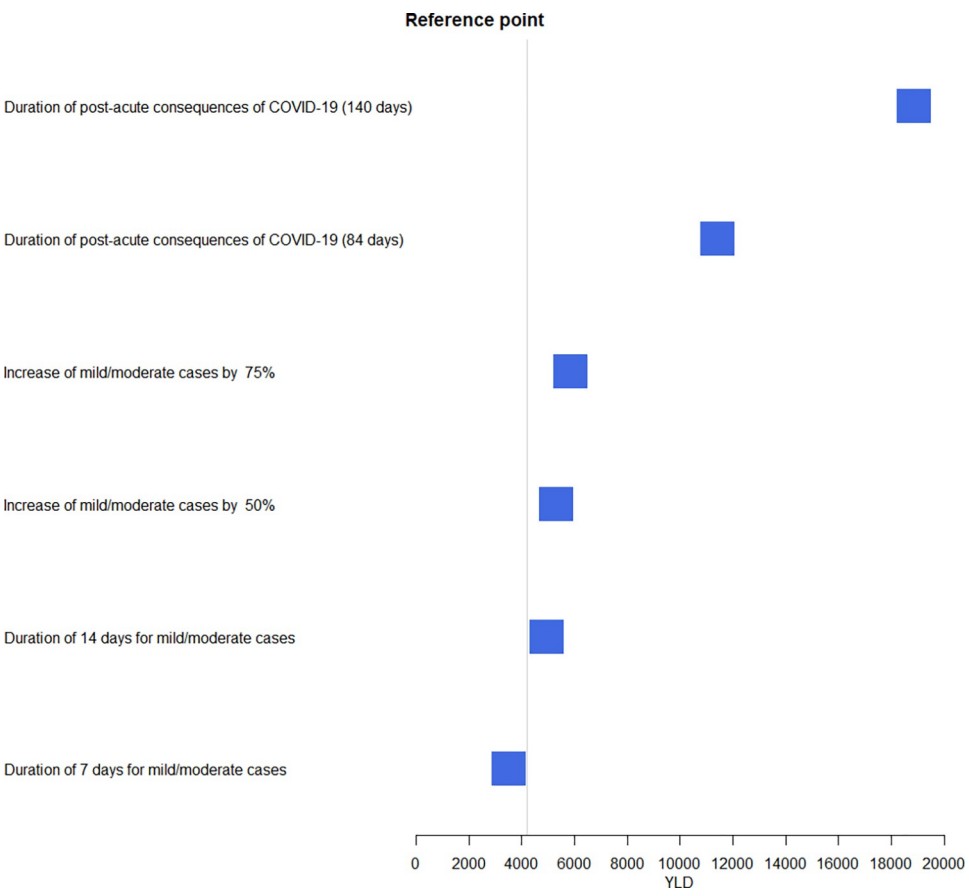

**Fig 5. Scenario analyses highlighting the impact of infections and the durations on YLD, France, 2020.**

(982 531) than in Germany (303 608) despite having a smaller population size. Both countries had different mortality data sources (i.e., surveillance data on notifiable diseases in Germany and causes of death statistics in France). France recorded a higher number of infections and

**Table 2. Scenario analyses highlighting the impact of number of infections and the durations on YLD and DALYs, France, 2020.**

| Health states | Parameters | YLD (UI 95%) | DALY (UI 95%) |
|---|---|---|---|
| **Acute symptomatic COVID-19 infections** | | | |
| | Reference | 4208 [4207–4210] | 986 740 [986 738–986 742] |
| | Increased mild/moderate cases by 50% | 5295 [5293–5298] | 987 827 [987 824–987 829] |
| | Increase mild/moderate cases by 75% | 5838 [5836–5842] | 988 370 [988 368–988 373] |
| | Duration of 7 days | 3483 [3482–3485] | 986 015 [986 14 – 986 16] |
| | Duration of 14 days | 4932 [4931–4935] | 987 465 [987 462–987 467] |
| **Post-acute consequences of COVID-19/Long COVID** | | | |
| | Duration of 28 days (Reference) | 3971 [3968–3974] | 986 503 [986 500–986 505] |
| | Duration of 84 days | 11 417 [11 409 – 11 424] | 993 948 [993 941–993 956] |
| | Duration of 140 days | 18 863 [18 850–18 875] | 1 001 394 [1 001 382–1 001 407] |

deaths in 2020 than Germany. The large difference in disease burden between these two countries could be explained by the fact that second wave was more or less over by the end of 2020 in France, whereas in Germany the peak of infections was not reached before January 2021. Most importantly, France used the GBD reference life table, whereas Germany used national life expectancy tables to estimate YLL. Scotland, Malta and France took into account the post-acute consequences of COVID-19. Scotland reported the highest contribution of post-acute consequences to YLD (76%) followed by Malta (60%) and then France (49%). It is important to take into account, which post-acute consequences were considered when interpreting these results. All studies estimated a large contribution of YLL to DALYs, compared to YLD. Overall differences in estimated COVID-19 disease burdens across countries may reflect differences in disease surveillance systems, data collection systems and hospital care as well as the heterogeneity in impact of COVID-19 during the first waves of the pandemic in 2020.

## Strengths and limitations

This is the first study that quantifies the disease burden due to COVID-19 using DALYs in France. We used all available relevant national data sources on COVID-19 infections, hospitalisations, and mortality to capture the direct impact of COVID-19 on population health. We performed a number of sensitivity analyses to measure the uncertainty around these estimates and highlighted their impact on resulting DALYs estimates. We used the GBD reference life table in order to facilitate the comparison of these results with other studies. Nevertheless, we acknowledged the choice of national or standard GBD life table is of intense debate [35]. Moreover, we adopted a standardized approach based on a consensus model that allowed to produce comparable estimates with other countries.

This study had limitations. *First,* we used the number of deaths with COVID-19 as an underlying cause of death to calculate the YLL estimates. Our analysis was based on approximately 80% of deaths certificates (coded) and 20% (not-coded and included the free text). These results may have an over estimated YLL, nevertheless, available data indicate that the extent of overestimation will likely be minimal. *Second,* to calculate the duration of severe and critical patients, we used a proxy indicator (i.e., the duration between the date of admission and the date of discharge) for each age group. These durations of severe and critical cases do not take into account whether patients have undergone further medical care, follow up and rehabilitation care, long-term care unit or psychiatry. *Third,* we observed a sizable variation in YLD estimates while increasing the number of acute symptomatic cases by 50% and 75%, despite a significant proportion of cases, remaining undiagnosed/untested [36]. However, these did not result in a large impact, as YLD has a relatively small contribution to DALY estimates. *Fourth,* YLD of post-acute consequences of COVID-19 were calculated using transition probabilities (1 in 7 cases of acute symptomatic infections). Currently, some French studies are ongoing to collect data on post-acute consequences, and at the time this study was conducted, evidence was scarce. Therefore, these results might be differ from the actual cases of post-acute consequences of COVID-19, so YLD estimates of post-acute consequences of COVID-19 should be interpreted with caution. *Fifth,* the maximum duration of long COVID is currently unknown but a subset of patients could suffer these consequences through the entire annual period of study and we still lack the data to assess these consequences. The duration of long COVID is a major determinant of YLD of post-acute consequences of COVID-19. Our disease model was based on a duration of 28 days, we used a threefold and fivefold (i.e., 84 days and 140 days) increase in duration of post-acute consequences as sensitivity analysis, to evaluate the impact of a much longer duration of post- acute consequences of COVID on YLD estimates. Nevertheless, the duration of post-acute consequences of COVID-19 remains

uncertain and more research is needed to draw conclusions on its duration. _Finally,_ SI-VIC data did not cover immigrants and homeless people, without social security numbers. These groups may account for at least 1% of the total population, and may have a much higher likelihood of suffering from poorer outcomes. Therefore, our DALY estimates may be a slight underestimated.

## Implications for public health

Estimates of DALYs, YLL and YLD due to COVID-19 in 2020 highlight the direct and immediate impact in terms of mortality and morbidity due to acute infections and post-acute consequences of COVID-19 among the French population. Among people aged younger than 70 years, YLL was much higher among men than women and these results have important implications for improving health care and prevention. YLD was higher among people aged younger than 70 years, a population group that constitutes an important contribution to the work force and professional activities. Consequently, this can lead to strong adverse economic and social impacts. In general, these results highlight important differences by sex. The concentration of disease burden among men already in middle age is very noticeable. This is an important aspect of the epidemiology of COVID-19, which has practical and political implications, and emphasizes the benefit of burden of disease indicators compared to classical epidemiological measures. We compared these estimates of COVID-19 DALYs with the top-ranking diseases and injuries reported by GBD 2019. COVID-19 ranked first (990 710 DALYs), followed by low back pain (927 416), ischemic heart disease (897 075), lung cancer (880 499), falls (785 307), stroke (630 048), Alzheimer's disease (582 922) and depressive disorders (511 467). Nevertheless, the country specific analyses had limited comparability with the GBD data due to several reasons including the use of local data sources, different case definitions, methodological choices, etc. This study highlighted the importance of data collection practices especially in context of more specific information on symptoms, durations, comorbidities and post-acute consequences of COVID-19 to follow-up these patients. In addition, this study could support to improve the comprehensiveness and the quality of data collected for COVID-19 or any other future epidemic. These estimates are useful for informing public health decision-makers regarding the COVID-19 burden among subgroups of the population who were the most affected by mortality and to target the high-risk sub-groups (people with comorbidities or with immunosuppressant diseases, elderly people) to limit the impact of COVID-19 on mortality estimates. Risk factors for severe COVID-19 and hospital admission, and death include older age, male sex, non-white ethnicity, being disabled, and pre-existing comorbidities (including obesity, cardiovascular disease, respiratory disease, and hypertension) [37].

It is important to evaluate the long term impact of the COVID-19 crisis on population health, such as the disruption of the utilisation of vital health care services [38], increased mental health problems [39] and risk of increase in health inequalities [40]. These are thought to be some of the main consequences of the COVID-19 pandemic. The current literature on post-acute COVID-19 syndrome based on clinical studies highlighted the organ-specific sequelae of COVID-19 survivors, especially the persistence of symptoms after discharge from the hospital at 60–120 days follow up [41]. For example, the most common symptoms persisted beyond/after acute COVID-19 infection were fatigue (up to ~64% at 60 days follow-up) [24, 42–46], dyspnea (42–66% at 60–100 days follow-up) [42, 43, 46, 47], chest pain (up to ~ 20% at 60 days follow-up) [42, 44] and sleep disturbance (24–31% at 90–120 days follow-up) [45, 46]. This evidence highlights the health care needs for patients with sequelae of COVID-19, such as fatigue, sleep disturbance, which are potentially not measureable in the data used (since they do not lead to hospital stays), therefore, the burden of long-COVID is underestimated. These

evidences suggest that developing integrated approaches across different disciplines for improved mental and physical health of survivors of COVID-19 in the long term are required [41]. Further research is needed to develop an updated consensus on defining the scope of long COVID in relation to the burden of disease calculations, especially in terms of what symptoms persisted after acute COVID-19 infections, their durations and the estimation of relevant disability weights. As perspective of this study, we propose to assess the impact of health inequalities within these estimates, and to update these estimates at subnational level. We recommend further research to compare the mortality estimates by the CépiDC with the WHO reported estimates that were published very timely during the pandemic. More research is required to measure the indirect impact of COVID-19 as a risk factor on the collateral damage to health care services, mental health, and increases in the incidence of non-communicable diseases.

## Conclusions

COVID-19 had a substantial impact on the population health in France in 2020. The majority of population health loss was due to mortality, especially among people aged 70 years and above. Overall, men had higher population health loss due to mortality and morbidity due to COVID-19 infections than women. Our analysis highlighted that the post-acute consequences of COVID-19 had a large contribution to the YLD component of the disease burden, even when we assume the shortest duration of 28 days, long COVID burden is large. This is still a new domain for research and more evidence is needed to understand the dynamics of the post-acute consequences of this infection in terms of persistence of certain symptoms, their durations and relevant disability weights, in relation to burden of disease calculations. Further research is recommended to assess the impact of health inequalities within these estimates and to measure the indirect impact of COVID-19 as a risk factor.

## Supporting information

**S1 Table. It is a doc. word file and describes the duration of hospitalization and hospitalization with intensive care by age group in 2020.**
(DOCX)

**S2 Table. It is a doc. word file and describes the results of scenario analyses highlighting the impact of number of infections and the durations on YLD and DALYs, by sex France, 2020.**
(DOCX)

**S3 Table. It is a doc. word file and describes YLD due to acute symptomatic COVID-19 infections in France in 2020 by age, sex and severity level.**
(DOCX)

**S4 Table. It is a doc. word file and describes the comparison of COVID-19 DALYs studies in European countries.**
(DOCX)

## Acknowledgments

The authors would like to acknowledge the technical networking support from COST Action CA18218 (European Burden of Disease Network: www.burden-eu.net: Under European Cooperation in Science and Technology). We thank Julie Figoni (Department of Infectious Diseases, Santé Publique France, Saint-Maurice, France) for her clinical expertise in dealing

with COVID-19 patients, Annabelle Lapostolle (Directions of Regions—Mayotte) for her inputs in study protocol.

## Author Contributions

**Conceptualization:** Romana Haneef, Christophe Bonaldi, Anne Gallay.

**Data curation:** Romana Haneef, Myriam Fayad, Anne Fouillet, Cécile Sommen.

**Formal analysis:** Romana Haneef, Myriam Fayad, Christophe Bonaldi.

**Investigation:** Romana Haneef.

**Methodology:** Romana Haneef, Myriam Fayad, Anne Fouillet, Cécile Sommen, Christophe Bonaldi, Grant M. A. Wyper, Sara Monteiro Pires, Brecht Devleesschauwer, Daniel Levy-Bruhl.

**Project administration:** Romana Haneef.

**Resources:** Romana Haneef.

**Supervision:** Romana Haneef, Anne Gallay.

**Validation:** Romana Haneef, Christophe Bonaldi, Grant M. A. Wyper, Sara Monteiro Pires, Brecht Devleesschauwer.

**Visualization:** Romana Haneef, Myriam Fayad.

**Writing – original draft:** Romana Haneef, Myriam Fayad, Anne Fouillet, Christophe Bonaldi, Grant M. A. Wyper, Sara Monteiro Pires, Brecht Devleesschauwer.

**Writing – review & editing:** Romana Haneef, Myriam Fayad, Anne Fouillet, Cécile Sommen, Christophe Bonaldi, Grant M. A. Wyper, Sara Monteiro Pires, Brecht Devleesschauwer, Antoine Rachas, Panayotis Constantinou, Daniel Levy-Bruhl, Nathalie Beltzer, Anne Gallay.

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
