## [Decision Letter · Decision Letter 0]

2 Jun 2022

PONE-D-22-11149Direct impact of COVID-19 by estimating disability-adjusted life years at national level in France in 2020PLOS ONE

Dear Dr. Haneef,

Thank you for submitting your manuscript to PLOS ONE. After careful consideration, we feel that it has merit but does not fully meet PLOS ONE’s publication criteria as it currently stands. Therefore, we invite you to submit a revised version of the manuscript that addresses the points raised during the review process.

Both reviewers agree that your manuscript requires major changes and provide constructive suggestions. Please address all of their comments and suggestions before resubmitting.==============================

We look forward to receiving your revised manuscript.

Kind regards,

Joël Mossong, PhD

Academic Editor

PLOS ONE

Journal Requirements:

2. Please clarify within the ethics statement whether the current study had authorization of the CNIL for the treatment of personal health data.

"I have read the journal's policy and the authors of this manuscript have the following competing interests: Romana Haneef is the first and corresponding author of this paper, is the section editor of “health information system” of “Archives of Public Health”. Brecht Devleesschauwer is the co-authors of this paper, is the editor of article collection on “burden of disease” of “Archives of Public Health. All other authors declare that they have no competing interests related to the work."

Reviewers' comments:

Reviewer's Responses to Questions

**Comments to the Author**

1. Is the manuscript technically sound, and do the data support the conclusions?

Reviewer #1: Partly

Reviewer #2: Partly

2. Has the statistical analysis been performed appropriately and rigorously? 

Reviewer #1: Yes

Reviewer #2: Yes

3. Have the authors made all data underlying the findings in their manuscript fully available?

Reviewer #1: No

Reviewer #2: Yes

4. Is the manuscript presented in an intelligible fashion and written in standard English?

Reviewer #1: Yes

Reviewer #2: Yes

5. Review Comments to the Author

Reviewer #1: General remark: the article is written in standard English as required but would nevertheless benefit from a professional proof reading.

General remark: DALY are mainly a comparative measure. While the present paper presents the 1st analysis of the burden of C-19 in France it remains limited to the comparison by age and gender. It shows, that the differences by sex are striking and the concentration of disease burden in men already in middle age is very noticeable. This is an important aspect of the epidemiology of C-19, which has practical and political implications, and illustrates the benefit of burden of disease indicators compared to classical epidemiological measures. However, in the paper more attention could be paid to these aspectsby highlighting these differences in the results section and in the discussion. So far, the scenario analyses (as a methodological by-product) as well as the absolute DALY, which in themselves are not very intuitive, take up comparatively large space.

In detail:

Line 70: It should be „November 2020” not “2021”

Line 117: Please state explicitly the case definition for the present analysis not only what is in the data: were all positive cases (PCR and rapid tests) included. Was a confirmatory 2nd rapid test required? Were there any exclusions defined (e.g. based on plausibility checks)?

Line 118: what is meant by “infra-national”

Lines 127/128: The whole concept of uncertainty should be explained in more detail. If one assumes a complete collection of data, there actually no uncertainty would be expected. Under what assumptions and with what methods was uncertainty calculated. The explanation that is given only refers to uncertainty intervals for YLD. Do uncertainty intervals for DALY only refer to the uncertainty in YLD? Is there no uncertainty assumed for YLL? If a concept of uncertainty is applied it should be said which aspects it covers and which aspects not.

Lines 140/141: It would be interesting to show the resulting severity distribution, maybe even by time and/or by age/sex.

Lines 160/161: The Australian study as well calculates durations from hospital data and ends up with somewhat different results compared to the French study (https://www.aihw.gov.au/getmedia/a69ee08a-857f-412b-b617-a29acb66a475/aihw-phe-287.pdf.aspx?inline=true). On the other hand, the Australian results compare quite well with the assumptions on durations made in the German study. Even if the impact on YLD is limited such aspects could be discussed in more detail.

Lines 177 et seq.: Please be a bit more explicit and detailed again on what definitions exist in the data, which case definitions were made for the analysis and possible exclusions due to data cleaning. Generally, there are three potential types of C-19 deaths: (i) dying with C-19 infection (but maybe has died in a car crash) (ii) dying due to C-19, where C-19 is one of several causes on the death certificate and (iii) deaths with C-19 as the main underlying cause of death. What definitions have been received here. As I understand it, the definition in the 80% and the 20% may not be quite the same? A time lag in the data is also mentioned here. How do the figures on deaths used here compare with those that were published very timely during the pandemic and reported to the WHO, for example? Are there major differences?

Lines 208-210: What is raised in the methods section should be presented under results. These aspects first come up in the discussion. Delete this in the methods section but as well briefly discuss the limited comparability of country specific analyses with GBD data.

Lines 248/249: Please stress a little bit the sex/gender differences here. Younger men account for a much higher share of YLL than younger women. This is an interesting phenomenon with many potential implications for prevention and health care.

Lines 272/273: It’s the other way round isn’t it? The share o acute infections within the age specific burden of C-19 is higher in younger age groups.

Line 278 et seq.: When presenting the scenario results you could give some information to what extend the share of YLD (in %) changes when introducing different scenarios (from 99% in the main analysis). This would be a bit more intuitive than just comparing absolute YLD.

Line 297 et seq.: The 1st paragraph of the discussion should be a summary of the main results. Here it is focused on the scenarios. However, this is not mainly a methodological study. Scenarios could be integrated later, e.g. in the limitations paragraph.

Lines 303/304: This conclusion is far too strong (also in the abstract). Although this is the variable with the largest effect in the scenario analyses, the impact on overall DALYs remains nevertheless very limited.

Lines 310-312: Germany deviates somewhat more from the consensus model compared to the other countries. The analyses were published before and did not make direct reference to the consensus model. The disease model covers somewhat different health states and did not consider Long COVID because evidence was scarce at that time.

Lines 324/325: I wonder to what extend these differences may also be due to different case definitions especially deaths. In some countries all cases dying with COVID-19 are reported, in others like Germany only cases having died due to C-19 were considered, in France, as far as I understood, only cases with C-19 as the main underlying cause were included (see commentary lines 177 et seq.). This should make a difference in total DALY that could be discussed here.

Line 321: Given the fact that Germany has the largest population it is a bit meaningless to state that it has the most infections.

Reviewer #2: Thank you for the opportunity to review ‘Direct impact of COVID-19 by estimating disability-adjusted life years at national level in France in 2020’. This well-written paper adds to the ongoing estimates of the relative importance of death, short-term, and long-term disability in determining total COVID-19 morbidity.

My major concern with this paper is the implausibly low duration of long-COVID, estimated from an analysis which in my view does not adequately distinguish censored subjects from those with events or either of these from missing data. Symptom duration was defined, roughly, as time from diagnosis to the first one-week period where neither symptoms nor hospitalization were reported; and those subjects who reported being seriously ill and then stopped logging were excluded from the analysis. Given that fatigue and brain fog are among the most common long-COVID symptoms, it is likely that the sickest patients more-often stopped reporting. Precisely the data which we would need to draw adequate conclusions, are missing.

I would prefer if the current publication relied on studies which actively contact patients, such as those reviewed in https://www.nature.com/articles/s41591-021-01283-z. These are likely to be much less biased by incomplete ascertainment, especially those such as Huang et al (cited in the current paper) with near-complete response; or Kim et al (https://bmcinfectdis.biomedcentral.com/articles/10.1186/s12879-022-07062-6). Even these are likely to underestimate duration somewhat, since postinfectious conditions tend to relapse and remit: a patient who is recovered at the time of followup may relapse again. This should be mentioned in the Discussion.

I also request that clinical manifestations other than long-COVID, such as myocarditis and psychiatric illness, be considered. These are described in Nalbandian et al, cited above. For simplicity and to avoid making the paper too long, I would be satisfied if their incidence rates were summed and the disability weights averaged to provide a single term for “all other conditions.”

Lastly, I request an extension of the literature review, comparing the current estimates with those others have found. The current paper may be the first to consider both in-hospital morbidity and long-COVID, in addition to death (many others explicitly exclude long-COVID, e.g. https://www.researchsquare.com/article/rs-1026794/v2) however it estimates a much higher share of YLD than https://www.jclinepi.com/article/S0895-4356(21)00339-5/fulltext . Please compare your findings to those of the latter paper, as well as any others you may find that compute DALY for long-COVID.

These issues are the only ones I have with the publication of the paper. If they are fully implemented I do not anticipate a second round of edits: the writing style does not require line edits. Please let me know if anything is unclear or I can be of further assistance.

6. PLOS authors have the option to publish the peer review history of their article (what does this mean?). If published, this will include your full peer review and any attached files.

Reviewer #1: No

Reviewer #2: No

---

## [Author Response · Author response to Decision Letter 0]

5 Jul 2022

Journal Requirements:

Response: Thank you. We have removed the physical address of the corresponding author from the title page and ensured that manuscript meets the PLOS ONE’s style requirements.

2. Please clarify within the ethics statement whether the current study had authorization of the CNIL for the treatment of personal health data.

Response: We had mentioned this in the main text of the manuscript (line 211) and have also clarified it within the ethics statement as follows:

“The study was based on aggregated and anonymous data from national mandatory database (SIVIC: article L3131-9-1 and R3131-10-1 code de la santé publique; SIDEP: article 11 loi n°2020-546 and article 8 and following décret n°2020-551). Furthermore, Santé publique France has been granted by French Law specific access to these national databases in order to carry out its mission of health monitoring and response to health crisis (article L. 1413-7 code de la santé publique). Therefore, no ethics approval or consent of participant was needed, as the processing made by Santé publique France is provided for by the Law.”

"I have read the journal's policy and the authors of this manuscript have the following competing interests: Romana Haneef is the first and corresponding author of this paper, is the section editor of “health information system” of “Archives of Public Health”. Brecht Devleesschauwer is the co-authors of this paper, is the editor of article collection on “burden of disease” of “Archives of Public Health”. All other authors declare that they have no competing interests related to the work."

Response: We have added the updated Competing Interests statement in the cover letter. 

Response to reviewers’ comments

Reviewer #1: 

General remark: the article is written in Standard English as required but would nevertheless benefit from a professional proof reading.

Response: Thank you very much. While we chose not to hire professional proof reading services, we made a thorough language check and revision of the manuscript.

General remark: DALY are mainly a comparative measure. While the present paper presents the 1st analysis of the burden of C-19 in France it remains limited to the comparison by age and gender. It shows that the differences by sex are striking and the concentration of disease burden in men already in middle age is very noticeable. This is an important aspect of the epidemiology of C-19, which has practical and political implications, and illustrates the benefit of burden of disease indicators compared to classical epidemiological measures. However, in the paper more attention could be paid to these aspects by highlighting these differences in the results section and in the discussion. So far, the scenario analyses (as a methodological by-product) as well as the absolute DALY, which in themselves are not very intuitive, take up comparatively large space. 

Response: Thank you very much for this comment. Indeed, we agree with you and we compared these estimates with other countries. Accordingly, we have added the following text in the result and discussion sections:

Results (line 265-267)

« The YLL among people [….] (Fig 1) and women accounted a higher share of YLL than men (37% vs 31%). On the other hand, among people younger than 70, men accounted for a much higher share of YLL than women (31% vs 19%). »

 Line 277

« Contrary to YLL, among people younger than 70, women accounted for a slightly higher share of YLD than men (69% vs 66%). »

Discussion (line 433)

« Among people younger than 70, YLL were much higher among men than women and these results have important implications for improving health care and prevention. In general, these results highlight the important differences by sex. The concentration of disease burden among men already in middle age is very noticeable. This is an important aspect of the epidemiology of COVID-19, which has practical and political implications, and emphasizes the benefit of burden of disease indicators compared to classical epidemiological measures. »

In detail: 

Line 70: It should be „November 2020” not “2021”

Response: This has been updated in the manuscript (line 67).

« November 2020 »

Line 117: Please state explicitly the case definition for the present analysis not only what is in the data: were all positive cases (PCR and rapid tests) included. Was a confirmatory 2nd rapid test required? Were there any exclusions defined (e.g. based on plausibility checks)?

Response: For this study, only positive cases with PCR or rapid/antigen tests were included. The same case definition as SI-DEP was used to identify the positive cases with PCR and rapid/antigen tests. If the first rapid/antigen test was positive and followed by a negative confirmatory PCR test, it was still considered as a positive case. 

We have added the following case definition for the present analysis and updated it in the manuscript file (Line 114):

« For this study, we considered all the cases who were tested positive (with PCR or antigen tests) once during a period of 60 days and were included in the present analysis. The case definition of positive cases was based on the SI-DEP database as 60 days […]. » 

Line 118: what is meant by “infra-national”

Response: We meant sub-national and is replaced by « infra-national » in the manuscript (Line 113). 

« Subnational »

Lines 127/128: The whole concept of uncertainty should be explained in more detail. If one assumes a complete collection of data, there actually no uncertainty would be expected. Under what assumptions and with what methods was uncertainty calculated. The explanation that is given only refers to uncertainty intervals for YLD. Do uncertainty intervals for DALY only refer to the uncertainty in YLD? Is there no uncertainty assumed for YLL? If a concept of uncertainty is applied it should be said which aspects it covers and which aspects not.

Response: Thank you very much for this comment. We have added an explanation under the section of « scenario analyses » (line 220-238) as follows and have deleted the sentence (line 125) to avoid confusion on uncertainty: 

«We performed a sensitivity analysis that corresponds to the variation of input parameters including the number of mild/moderate cases and the duration according to the scenario. 

During the epidemic, many cases with mild and moderate symptoms may have remained undiagnosed/untested and not registered in the SI-DEP database. At the time of submission of this study, the true incidence of COVID-19 for the whole year of 2020 in France was not available. Therefore, to evaluate the impact of underreporting underestimation of the burden of disease, we varied the proportion of reported cases by increasing them by 50% and 75%, respectively. 

The duration of infection for a mild/moderate case was not recorded in SI-DEP. The transmission of the virus is different across the whole population, influenced by environmental and behavioral factors, and varied from person to person. Therefore, the duration of mild/moderate cases remains uncertain. To take into account the uncertainty around this parameter, we varied the duration from 7 days to 14 days. Similarly, the duration of post-acute consequences of COVID-19 was based on transition probabilities calculated in a previous study. At the time this study was performed, no data on duration were available.

Therefore, we varied the number of mild/moderate cases and the duration to quantify the uncertainty intervals of YLD as described below. We did not apply this concept to the YLL as the national mortality database (i.e., CépiDc) is considered a robust data source with good quality of data and no data adjustment was applied. »

Lines 140/141: It would be interesting to show the resulting severity distribution, maybe even by time and/or by age/sex.

Response: As suggested, we have mentioned in the method section and added a figure with a brief description in the results section as follows:

Method (health states) (line 139)

« We also described the severity distribution of these health states by age and sex. »

Results (line 283)

« The severity distribution of YLD varied among men and women (Fig 3). Overall, the proportion of YLD due to mild/moderate cases was 60% among women whereas 44% among men. Among women younger than 70, they shared a larger proportion of YLD due to mild/moderate cases than women aged 70 and above (89% vs 11%). The proportion of YLD due to severe cases was almost similar in both sexes (20% women vs 19% men). The YLD due to critical cases was higher among men (37%) than women (20%). » 

Fig 3 (line 290): Years of Life Lived due to acute symptomatic COVID-19 infections in France in 2020 by age, sex and severity level »

Lines 160/161: The Australian study as well calculates durations from hospital data and ends up with somewhat different results compared to the French study (https://www.aihw.gov.au/getmedia/a69ee08a-857f-412b-b617-a29acb66a475/aihw-phe-287.pdf.aspx?inline=true). On the other hand, the Australian results compare quite well with the assumptions on durations made in the German study. Even if the impact on YLD is limited such aspects could be discussed in more detail.

Response: Thank you for this comment. We have cited the « Australian study » in the list of countries who calculated the DALYs estimates under the discussion section (line 341):

« Other than European countries, Australia has also calculated DALYs estimates using the Burden of Disease consensus disease model for COVID -19 (AIHW). »

The following text on different assumptions for duration in relation to the results has been added under the discussion section (line 347): 

« For example, Australia applied the same assumption for the duration of hospitalized and critical cases (i.e., average length of stay) as France but had different results. The hospital stay among people aged 80 and over was longer in France than in Australia (i.e., 20 days vs 12 days). These differences highlight that patients were critically ill and were required longer time in intensive care in France than in Australia. The heterogeneity in health care in the hospitals and health insurance system might affect the length of stay in the hospital, consequently, which can influence the YLD estimates. Overall, the length of stay in hospital increased with age in both countries. » 

Lines 177 et seq.: Please be a bit more explicit and detailed again on what definitions exist in the data, which case definitions were made for the analysis and possible exclusions due to data cleaning. Generally, there are three potential types of C-19 deaths: (i) dying with C-19 infection (but maybe has died in a car crash) (ii) dying due to C-19, where C-19 is one of several causes on the death certificate and (iii) deaths with C-19 as the main underlying cause of death. What definitions have been received here. 

Response: Precisely, we included those deaths with COVID-19 as the main underlying causes of death in our analysis. This was already reported in the text under the method section and we updated it more explicitly as requested:

Method (line 187)

« The underlying causes of death are selected using an automatic system (IRIS software [21]) for coding and prioritizing multiple cause of death. »

Line 197

« COVID-19 deaths were defined as deaths reported with COVID-19 as the main underlying cause of death according to the WHO recommendations. » 

Line 201

« In the main analysis, we included those deaths with COVID-19 as the main underlying cause of death. »

As I understand it, the definition in the 80% and the 20% may not be quite the same? A time lag in the data is also mentioned here. 

Response: We already mentioned in the manuscript that the 80% deaths are based on underlying cause of death and 20% include deaths reported as the free text with a mention of COVID-19 (line 192 – 195). The COVID-19 death is the common word and the algorithm applied to extract COVID-19 terms from the free-text section of the death certificate, has a high performance of 93%. 

How do the figures on deaths used here compare with those that were published very timely during the pandemic and reported to the WHO, for example? Are there major differences?

Response: For this study, we did not formally compare this data with the WHO reported ones. It is likely that the mortality data used in the study was provided by CépiDC, are more reliable than those published by WHO, which might have counted the COVID-19 death numbers as cumulative number of deaths from all causes over time. Nevertheless, we compared mortality estimates with other countries who had performed DALYs estimations. We have added this point as further research in the discussion section as follows: 

Discussion (line 463)

« We recommend further research to compare the mortality estimates by the CépiDC with the WHO reported ones that were published very timely during the pandemic. »

Lines 208-210: What is raised in the methods section should be presented under results. These aspects first come up in the discussion. Delete this in the methods section but as well briefly discuss the limited comparability of country specific analyses with GBD data.

Response: As requested, we have deleted this from the method section and mentioned following text in the discussion section (line 443): 

« Nevertheless, the country specific analyses had limited comparability with the GBD data due to several reasons including the use of local data sources, different case definitions, methodological choices, etc. ». 

Lines 248/249: Please stress a little bit the sex/gender differences here. Younger men account for a much higher share of YLL than younger women. This is an interesting phenomenon with many potential implications for prevention and health care.

Response: Thank you for this comment. We have added the following text in the result and discussion sections based on your first comment:

Results (line 265-267)

« The YLL among people [….] (Fig 1) and women accounted a higher share of YLL than men (37% vs 31%). On the other hand, among people younger than 70, men accounted for a much higher share of YLL than women (31% vs 19%). »

(Line 277)

« Contrary to YLL, among people younger than 70, women accounted for a slightly higher share of YLD than men (69% vs 66%). »

Discussion (line 433)

« Among people younger than 70, YLL was much higher among men than women and these results have important implications for improving health care and prevention. In general, these results highlight the important differences by sex. The concentration of disease burden among men already in middle age is very noticeable. This is an important aspect of the epidemiology of COVID-19, which has practical and political implications, and emphasizes the benefit of burden of disease indicators compared to classical epidemiological measures. »

Lines 272/273: It’s the other way round isn’t it? The share of acute infections within the age specific burden of C-19 is higher in younger age groups.

Response: Indeed, the share of YLD due to acute infections within the age specific burden of COVID-19 is higher in younger age groups (67% vs 33%) [line 276-277] but for DALYs, COVID-19 burden due to acute symptomatic infections is higher among people aged 70 and above then among people under 70 years (74% vs 26%) [line 302]. 

Line 278 et seq.: When presenting the scenario results you could give some information to what extend the share of YLD (in %) changes when introducing different scenarios (from 99% in the main analysis). This would be a bit more intuitive than just comparing absolute YLD.

Response: Thank you for this comment to further improve the manuscript. We have added the % increase in the YLD estimates by varying different scenarios as follows under the results section of « scenario analyses »:

Results (scenario analyses) (line 309 – 317) 

« First, by increasing the number of positive cases by 50 % or 75%, it led to an increase of 26% (5295) and 39% (5838) of YLD, respectively, as compared to the main results of YLD (4208). In turn, 987 827 and 988 370 DALYs were estimated, respectively, as compared to the main results of DALYs (986 740). Second, by varying the duration from 7 days to 14 days, it led to an increase of 17% for both durations with a total of 3484 and 4933 YLDs, respectively. Third, by varying the duration for post-acute consequences of COVID-19 cases to 84 days, it led to an increase of 40% with a total of 11 417 YLDs […]. A detailed analysis by sex is reported in S2 Table. » 

Line 297 et seq.: The 1st paragraph of the discussion should be a summary of the main results. Here it is focused on the scenarios. However, this is not mainly a methodological study. Scenarios could be integrated later, e.g. in the limitations paragraph.

Response: We agree with this comment and as requested, we have moved the text related to the scenario results from the main results to the limitation section (as limitation 3 - line 393). 

Lines 303/304: This conclusion is far too strong (also in the abstract). Although this is the variable with the largest effect in the scenario analyses, the impact on overall DALYs remains nevertheless very limited.

Response: We have updated the tone of this sentence as follows:

Abstract (conclusions) (line 50)

« Post-acute consequences of COVID-19 have a higher contribution to the YLD component of the burden than acute symptomatic infections [...] »

Discussion and conclusions (line 332 and 472)

« Our analysis highlighted that post-acute consequences of COVID-19 have a higher contribution to the YLD component of the burden than acute symptomatic infections and thus […]. »

Lines 310-312: Germany deviates somewhat more from the consensus model compared to the other countries. The analyses were published before and did not make direct reference to the consensus model. The disease model covers somewhat different health states and did not consider Long COVID because evidence was scarce at that time.

Response: We agree that at the time of publication of German study, very limited evidence was available. We have deleted the relevant text and added the following text in the discussion section (line 366). 

«The German study was published very early during the pandemic. The consensus disease model covered health states of acute symptomatic COVID-19 infections and did not consider post-acute consequences of COVID-19 due to scarce evidence at that time. »

Lines 324/325: I wonder to what extend these differences may also be due to different case definitions especially deaths. In some countries all cases dying with COVID-19 are reported, in others like Germany only cases having died due to C-19 were considered, in France, as far as I understood, only cases with C-19 as the main underlying cause were included (see commentary lines 177 et seq.). This should make a difference in total DALY that could be discussed here.

Response: Thank you for this interesting comment. It is important to take into account the variability in data sources and the availability of official mortality estimates. As, it has been discussed in the German study that “Moreover, persons with COVID-19 can die from multiple causes; the recording of only the main cause of death of the list of underlying causes, as is done in Germany, is problematic (Wengler A et al, 2019 and Wengler A et la, 2020). However, future analyses based, e.g., on cause-of-death statistics for 2020 will be able to reveal whether COVID-19 was given as the official cause of death in numbers comparable to the reported fatal cases of COVID-19”. 

Nevertheless, we have added the word « different case definitions especially for death » (line 345) and the following sentence (line 356):

« In German study, only deaths due to COVID-19 were considered whereas, in France, the deaths with COVID-19 as the main underlying cause were included in the main analysis. This difference in case definitions of COVID-19 death might influence the number of deaths counted that can have an impact on YLLs and then total DALYs. For example, in France YLL counted higher (982 531 YLL: death with COVID-19 as underlying cause of death) than in Germany (303 608 YLL: death due to COVID). Nevertheless, it is important to take into account that both countries have different mortality data sources and it is possible that the official estimates of the causes of death were not available in Germany at the time study was conducted and it underestimated the YLL. »

Line 321: Given the fact that Germany has the largest population it is a bit meaningless to state that it has the most infections.

Response: We have deleted the related sentence in text. 

Reviewer #2: Thank you for the opportunity to review ‘Direct impact of COVID-19 by estimating disability-adjusted life years at national level in France in 2020’. This well-written paper adds to the ongoing estimates of the relative importance of death, short-term, and long-term disability in determining total COVID-19 morbidity.

Response: Thank you very much for this compliment.

My major concern with this paper is the implausibly low duration of long-COVID, estimated from an analysis, which in my view does not adequately distinguish censored subjects from those with events or either of these from missing data. Symptom duration was defined, roughly, as time from diagnosis to the first one-week period where neither symptoms nor hospitalization were reported; and those subjects who reported being seriously ill and then stopped logging were excluded from the analysis. Given that fatigue and brain fog are among the most common long-COVID symptoms, it is likely that the sickest patients more-often stopped reporting. Precisely the data which we would need to draw adequate conclusions, are missing.

Response: Thank you for highlighting this point. We fully agree with you that precise data to draw adequate conclusions on long-COVID are missing and more evidence is needed. Currently, many studies are ongoing and in near future, their results would support to rule out various aspects of long-COVID. We have already mentioned it in the method section (line 162-165) the assumption of calculating the cases of long COVID, is based on previously calculated transition probabilities (Sudre C H et al, 2021). 

We have added the following text in the method section and updated it the discussion section as well:

Method (line 163)

« At the time this study was conducted, some studies were ongoing to collect the data on post-consequences of COVID-19 in France. » 

Discussion (line 404)

« Currently, many studies are ongoing to collect the data on post-acute consequences and at the time this study was conducted, scarce evidence was available. » 

I would prefer if the current publication relied on studies, which actively contact patients, such as those reviewed in https://www.nature.com/articles/s41591-021-01283-z. These are likely to be much less biased by incomplete ascertainment, especially those such as Huang et al (cited in the current paper) with near-complete response; or Kim et al (https://bmcinfectdis.biomedcentral.com/articles/10.1186/s12879-022-07062-6). Even these are likely to underestimate duration somewhat, since post infectious conditions tend to relapse and remit: a patient who is recovered at the time of follow up may relapse again. This should be mentioned in the Discussion.

Response: Thank you very much for this important point. We have cited following studies reported in the review and have mentioned this point in the discussion section (line 409):

« Nalbandian A. et al, has provided a comprehensive review of the current literature on post-acute COVID-19, its pathophysiology, its organ-specific sequelae and relevant considerations for the multidisciplinary care of COVID-19 survivors. Several studies have shown that […] (Carfi et al, Carvalho-Schneider et al, Chopra et al, Arnold et al, Moreno-Pérez et al, Garrigues et al). Even these studies are likely to underestimate duration somewhat, since post-acute infectious conditions tend to relapse and remit: a patient who is recovered at the time of follow up may relapse again. Therefore, these results underestimate the contribution of post-acute consequences of COVID-19 to YLD. »

I also request that clinical manifestations other than long-COVID, such as myocarditis and psychiatric illness, be considered. These are described in Nalbandian et al, cited above. For simplicity and to avoid making the paper too long, I would be satisfied if their incidence rates were summed and the disability weights averaged to provide a single term for “all other conditions.” 

Response: Thank you for this comment and the suggestion. 

« Summing up the incidence rates and the disability weights averaged to provide a single term for ‘all other conditions’ », this would be interesting. The main focus of this paper was to calculate the burden of COVID-19 and we briefly mentioned the symptoms of long COVID reported in published literature. We did not plan to describe the clinical manifestations other than long-COVID as it was not part of the European Burden of Disease Network consensus disease model for COVID-19. Moreover, due to lack of data and resources, we were not able to perform this analysis. Nevertheless, we mentioned this point as one of the limitations of this study (limitation 6) and also recommend further research to evaluate the impact of “all other conditions” as multimorbidity and comorbidity on DALYs estimates as follows: 

Discussion (line 417) 

« Sixth, for this study, we did not take into account the multimorbidity and comorbidity parameters in the estimation of DALYs due to the inconsistent nature of COVID-19 in presence of different chronic conditions and lack of precise data. Moreover, this aspect is currently debated and may introduce some ethical issues whose discussion is beyond the scope of this paper. Nevertheless, we recommend further research to explore the impact of multimorbidity and comorbidity on DALYs estimates. » 

Lastly, I request an extension of the literature review, comparing the current estimates with those others have found. The current paper may be the first to consider both in-hospital morbidity and long-COVID, in addition to death (many others explicitly exclude long-COVID, e.g. https://www.researchsquare.com/article/rs-1026794/v2) however it estimates a much higher share of YLD than https://www.jclinepi.com/article/S0895-4356(21)00339-5/fulltext . Please compare your findings to those of the latter paper, as well as any others you may find that compute DALY for long-COVID.

Response: Thank you for this comment. We have cited the relevant articles and the following text in the discussion section (line 370):

« Scotland reported the highest contribution of post-acute consequences to YLD (76%) followed by Malta (60%) and then France (49%). This highlights the large impact on population health. It is important to take into account what kind of post-acute consequences were considered, which might explain the heterogeneity in these results. Nevertheless, more evidence is needed to draw conclusions on its impact on the population (Maia P. Smith, 2022). »

These issues are the only ones I have with the publication of the paper. If they are fully implemented I do not anticipate a second round of edits: the writing style does not require line edits. Please let me know if anything is unclear or I can be of further assistance.

Response : Thank you

---

## [Decision Letter · Decision Letter 1]

1 Aug 2022

PONE-D-22-11149R1Direct impact of COVID-19 by estimating disability-adjusted life years at national level in France in 2020PLOS ONE

Dear Dr. Haneef,

Thank you for submitting your manuscript to PLOS ONE. After careful consideration, we feel that it has merit but does not fully meet PLOS ONE’s publication criteria as it currently stands. Therefore, we invite you to submit a revised version of the manuscript that addresses the points raised during the review process.

ACADEMIC EDITOR Both reviewers find that there are still some major issues which need to be addressed before the manuscript can be considered for publication. Please address all of their comments.

We look forward to receiving your revised manuscript.

Kind regards,

Joël Mossong, PhD

Academic Editor

PLOS ONE

Reviewers' comments:

Reviewer's Responses to Questions

**Comments to the Author**

1. If the authors have adequately addressed your comments raised in a previous round of review and you feel that this manuscript is now acceptable for publication, you may indicate that here to bypass the “Comments to the Author” section, enter your conflict of interest statement in the “Confidential to Editor” section, and submit your "Accept" recommendation.

Reviewer #1: All comments have been addressed

Reviewer #2: (No Response)

2. Is the manuscript technically sound, and do the data support the conclusions?

Reviewer #1: Yes

Reviewer #2: Partly

3. Has the statistical analysis been performed appropriately and rigorously? 

Reviewer #1: Yes

Reviewer #2: No

4. Have the authors made all data underlying the findings in their manuscript fully available?

Reviewer #1: Yes

Reviewer #2: Yes

5. Is the manuscript presented in an intelligible fashion and written in standard English?

Reviewer #1: Yes

Reviewer #2: Yes

6. Review Comments to the Author

Reviewer #1: General remark: I would still suggest having checked the text - at least some parts of the paper, especially the newly introduced parts of the results section - by a native speaker in order to make sure that what you want to say is actually understood in the same way. Some expressions still seem to be a bit misleading to me.

219 et seq The description of the scenario analyses contains some repetitions and could be shortened.

265 et seq “On the other hand, among people younger than 70, men accounted for a much higher share of YLL than women (31% vs 19%).” Maybe better: “On the other hand, men younger than 70 accounted for a much higher share of YLL in men (31%) than women of the same age for the share of YLL in women (19%).”

283 et seq While Fig 3 is a very interesting presentation of results (YLD by severity) it does not show the severity distribution as is claimed in the text. Severity distribution is defined as the distribution of diseased cases (not YLD) across the levels of severity (expressed in total numbers and/or percentages). Severity distributions could be presented in a supplementary table in order to make the data/methods more transparent.

302 et seq The sentence “The percentage of DALYs due to acute symptomatic infections were higher among people aged 70 and above than among people under 70 years (74% vs 26%).” is still a bit confusing. What do you mean by DALY due to acute infections? Is this YLD? If yes that it should be: “People aged 70 and above account for 74% of the total YLD” and so on. Put differently, the share of YLD in DALY should be higher in younger people compared to the elderly because only very few younger people die from C-19.

308 et seq In line 300 it is said: “We observed that 99% of DALYs were due to mortality (982 531 YLL) and only 1% was due to morbidity (8179 YLD).” How does this distribution changes in the scenario analyses, especially when introducing longer durations for long COVID? It is frequently argued that the high share of YLL within DALY would be strongly diminished when adequately considering long COVID.

355 It should say: “…whereas Germany only used national life tables”.

356 -364 I would suggest to delete or change this. The case definition in France and Germany is quite similar what differs is the data source (surveillance data on notifiable diseases in Germany, CoD statistics in France). However, if you want to explain the huge differences in DALY I would suggest mainly the following points: (i) much more infections in France (ii) much more deaths in France (iii) use of GBD life tables in France. (i) and (ii) can in part be explained by the fact that the 2nd wave was more or less over by the end of 2020 in France but had its peak not before January 2021 in Germany.

367 et seq Anayway, the impact of YLD on DALY remains very limited and the issue if or how long COVID is considered cannot explain the main differences in DALY between countries. An explanation should always put the focus on the mortality part. This should be clearly stated somewhere. See line 371 In fact the impact on population health is limited compared to mortality.

Reviewer #2: Authors did not implement the additional analyses I requested: instead they added a few sentences saying that they hadn't done so. They easily could have done those analyses with existing data.

Without those additional analyses the paper substantially misrepresents the situation and has potential to mislead policy if published as it stands. It could most charitably be described as incomplete / biased.

Please make the changes I requested in the previous round of edits, especially the sensitivity analysis using numbers for long-COVID prevalence obtained from contacting patients.

7. PLOS authors have the option to publish the peer review history of their article (what does this mean?). If published, this will include your full peer review and any attached files.

Reviewer #1: No

Reviewer #2: No

---

## [Author Response · Author response to Decision Letter 1]

7 Oct 2022

Response to reviewers’ comments

Reviewer #1: 

General remark: I would still suggest having checked the text - at least some parts of the paper, especially the newly introduced parts of the results section - by a native speaker in order to make sure that what you want to say is actually understood in the same way. Some expressions still seem to be a bit misleading to me.

Response: Thank you. The manuscript has been re-reviewed by a native English speaker, one of the co-authors of this manuscript, Grant Wyper (Place and Wellbeing Directorate, Public Health Scotland, Glasgow, Scotland, United Kingdom). 

219 et seq The description of the scenario analyses contains some repetitions and could be shortened.

Response: It is updated.

265 et seq “On the other hand, among people younger than 70, men accounted for a much higher share of YLL than women (31% vs 19%).” Maybe better: “On the other hand, men younger than 70 accounted for a much higher share of YLL in men (31%) than women of the same age for the share of YLL in women (19%).”

Response: This has been updated as follows (line 256) : 

« On the other hand, men younger than 70 years accounted for a much higher share of YLL (31% of 559 784) than women of the same age (19% of 422 747).”

283 et seq

While Fig 3 is a very interesting presentation of results (YLD by severity) it does not show the severity distribution as is claimed in the text. Severity distribution is defined as the distribution of diseased cases (not YLD) across the levels of severity (expressed in total numbers and/or percentages). Severity distributions could be presented in a supplementary table in order to make the data/methods more transparent.

Response: Thanks for this precision. Fig 3 describes the YLD across three levels of severity by age and sex. The text is updated in the method section as follows (line 134):

« These health states represent the different severity levels considered for COVID-19 cases. » 

« We also described the YLD estimates among males and females according to mild/moderate, severe and critical cases. » (line 175)

We have added a supplementary table as S3 Table, describing the YLD by severity level to make data more transparent.

302 et seq 

The sentence “The percentage of DALYs due to acute symptomatic infections were higher among people aged 70 and above than among people under 70 years (74% vs 26%).” is still a bit confusing. What do you mean by DALY due to acute infections? Is this YLD? If yes that it should be: “People aged 70 and above account for 74% of the total YLD” and so on. Put differently, the share of YLD in DALY should be higher in younger people compared to the elderly because only very few younger people die from C-19.

Response : The DALYs due to acute symptomatic infections only take into account the YLD of mild/moderate, severe and critical cases (i.e., 4208). The DALYs due to acute infections do not take into account the YLD due to long COVID (i.e., 3971). The suggested YLD interpretation has already reported in YLD result section.

We have clarified this phrase to avoid any confusion as follows (line 298-302) :

« The percentage of DALYs due to acute symptomatic infections (i.e., including only YLD of mild/moderate, severe and critical cases [4208] and YLL estimates) were higher among men than women (57% vs 43%). People aged 70 years and above had a higher DALY estimates compared to people under 70 years (728 688 vs 258 051). (Fig 4) »

308 et seq 

In line 300 it is said: “We observed that 99% of DALYs were due to mortality (982 531 YLL) and only 1% was due to morbidity (8179 YLD).” How does this distribution changes in the scenario analyses, especially when introducing longer durations for long COVID? It is frequently argued that the high share of YLL within DALY would be strongly diminished when adequately considering long COVID.

Response : The contribution of mortality remains high even after introducing the long duration of COVID-19 to 140 days as follows : 

DALY with 140 days duration (1 001 394 = 100%) = YLL (982 531 = 98%) + YLD (18 863 = 2%).

Therefore, we have updated this argument after considering the results of current additional sensitivity analysis of duration of long COVID throughout the manuscript.

355 It should say: “…whereas Germany only used national life tables”.

Response : It is updated as proposed (line 363). 

356 -364 

I would suggest to delete or change this. The case definition in France and Germany is quite similar what differs is the data source (surveillance data on notifiable diseases in Germany, CoD statistics in France). However, if you want to explain the huge differences in DALY I would suggest mainly the following points: (i) much more infections in France (ii) much more deaths in France (iii) use of GBD life tables in France. (i) and (ii) can in part be explained by the fact that the 2nd wave was more or less over by the end of 2020 in France but had its peak not before January 2021 in Germany.

Response : Thank you very much for this input. We have updated the text as suggested. 

Discussion (line 356-364)

« In France, YLL was higher (982 531) than in Germany (303 608) despite having a smaller population size. Both countries had different mortality data sources (i.e., surveillance data on notifiable diseases in Germany and causes of death statistics in France). France recorded a higher number of infections and deaths in 2020 than in Germany. The large difference in disease burden between these two countries could be explained by the fact that second wave was more or less over by the end of 2020 in France, whereas in Germany, the peak of infections was not reached before January 2021. Most importantly, France used the GBD reference life table, whereas Germany used national life expectancy tables to estimate YLL. »

367 et seq 

Anyway, the impact of YLD on DALY remains very limited and the issue if or how long COVID is considered cannot explain the main differences in DALY between countries. An explanation should always put the focus on the mortality part. This should be clearly stated somewhere. 

Response : We agreed and have added the following text :

Discussion (line 366-372)

“It is important to take into account, which post-acute consequences were considered when interpreting these results. All studies estimated a large contribution of YLL to DALYs, compared to YLD. Overall differences in estimated COVID-19 disease burdens across countries may reflect differences in disease surveillance systems, data collection systems and hospital care as well as the heterogeneity in impact of COVID-19 during the first waves of the pandemic in 2020.”

See line 371 In fact the impact on population health is limited compared to mortality.

Response: We agreed that mortality has much strong impact than morbidity even after considering the long duration of COVID long.

We have deleted this phrase from the abstract and discussion sections. 

Reviewer #2: 

Comment of first round:

My major concern with this paper is the implausibly low duration of long-COVID, estimated from an analysis, which in my view does not adequately distinguish censored subjects from those with events or either of these from missing data. Symptom duration was defined, roughly, as time from diagnosis to the first one-week period where neither symptoms nor hospitalization were reported; and those subjects who reported being seriously ill and then stopped logging were excluded from the analysis. Given that fatigue and brain fog are among the most common long-COVID symptoms, it is likely that the sickest patients more-often stopped reporting. Precisely the data, which we would need to draw adequate conclusions, are missing.

Response : We fully agree with the low duration of long-COVID and integrated an additional sensitivity analysis with a five-fold increase in baseline duration used for long-COVID (i.e., 28*5 = 140 days). This has been added in the method, results and discussion sections as follows :

Method – Scenario analyses (line 238) 

« Therefore, we assumed two scenarios, one with threefold (i.e., 84 days) and another one with a fivefold (i.e., 140 days) increase in duration of post-acute consequences. The choice of 84 and 140 days was based on published literature (Moreno-Pérez et al, Huang et al) and experts’inputs . »

Results- Scenario analyses (line 313)

« Third, varying […] to 84 days and 140 days, it led to an increase of 188% […] and 375% with a total of 18 863 YLD, respectively. The duration of 140 days had the highest effect on DALYs (i.e., 1 001 394).

Fig 5 is updated accordingly.

Discussion- Strength and limitations (line 400-405)

« Fifth, we […], using a fivefold increase in duration of post-acute consequences (i.e., 140 days) […]. Nevertheless, the duration of post-acute consequences of COVID remains uncertain and more research is needed to draw conclusions on its duration. » 

Supporting information-S2 Table is updated accordingly.

Comment of first round

I would prefer if the current publication relied on studies, which actively contact patients, such as those reviewed in https://www.nature.com/articles/s41591-021-01283-z. These are likely to be much less biased by incomplete ascertainment, especially those such as Huang et al (cited in the current paper) with near-complete response; or Kim et al (https://bmcinfectdis.biomedcentral.com/articles/10.1186/s12879-022-07062-6). Even these are likely to underestimate duration somewhat, since post infectious conditions tend to relapse and remit: a patient who is recovered at the time of follow up may relapse again. This should be mentioned in the Discussion.

Response

We have already cited this review in the first round of revisions and have mentioned the underestimation of the duration of long COVID-19. 

Now we updated the text and added a more detailed paragraph in the discussion section on what different studies have found in the context of long-COVID as follows : 

Discussion – Implications for public health 

Line 442-454

« The current literature on post-acute COVID-19 syndrome based on clinical studies highlighted the organ-specific sequelae of COVID-19 survivors, especially the persistence of symptoms after discharge from the hospital at 60 – 120 days follow up (Nalbandiean et al). For example, the most common symptoms persisted beyond/after acute COVID-19 infection were fatigue (upto ~64% at 60 days followup)(Carfi et al, Haplin et al, Carvalho-Schneider et al, Arnold et al, Moreno-Pérez et al, Garrigues et al) , dyspnea (42-66% at 60-100 days followup) (Carfi et al, Chopra et al, Halpin et al, Garrigues et al), chest pain (upto ~ 20% at 60 days followup)(Carfi et al, Carvalho-Schneider et al) and sleep disturbance (24 -31% at 90-120 days followup)(Arnold et al, Garrigues et al). This evidence highlights the health care needs for patients with sequelae of COVID-19, such as fatigue, sleep disturbance, which are potentially not measurable in the data used (since they do not lead to hospital stays), and therefore, the burden of long-COVID is underestimated. These evidences suggest that developing integrated approaches across different disciplines for improved mental and physical health of survivors of COVID-19 in the long term are required (Nalbandiean A et al).” 

Comments of first round:

I also request that clinical manifestations other than long-COVID, such as myocarditis and psychiatric illness, be considered. These are described in Nalbandian et al, cited above. For simplicity and to avoid making the paper too long, I would be satisfied if their incidence rates were summed and the disability weights averaged to provide a single term for “all other conditions.” 

Response

We have discussed this point with our team and still consider that this additional analysis creates difficulties due to the following reasons:

First, our estimates were calculated based on the consensus model developed by the European Burden of Disease experts (validated & published at SSPH+ | Burden of Disease Methods: A Guide to Calculate COVID-19 Disability-Adjusted Life Years (ssph-journal.org)). This model does not take into account the “all other conditions” in the context of long COVID. This would require a consensus on defining the scope of long COVID in relation to the burden of disease calculations, in terms of what symptoms persisted after acute COVID-19 and the relevant disability weights to be used. Currently, there is no “averaged disability weight” that could take into account “all other conditions” proposed by the reviewer. While we are aware that the team that developed the consensus model, is working on updating the relevant parameters for long-COVID, at this stage it is not available yet. We have highlighted this point as a recommendation in the discussion and the main conclusion sections

Discussion (line 454) 

“Further research is needed to develop an updated consensus on defining the scope of long COVID in relation to the burden of disease calculations, especially in terms of what symptoms persisted after acute COVID-19 infections, their durations and the estimation of relevant disability weights.” 

Conclusions (line 470)

“This is still a new domain for research and more evidence is needed to understand the dynamics of the post-acute consequences of this infection in terms of persistence of certain symptoms, their durations and relevant disability weights, in relation to burden of disease calculations.”

Second, we think summing up the incidence rates of the post-acute COVID-19 studies reported in Nalbandian et al review (https://www.nature.com/articles/s41591-021-01283-z) (suggested by the reviewer 2) is not directly applicable to the French context. For example, these estimates are high by design. Studies included those patients who were hospitalized and were followed up, which arguably could increase the need for a control group (11% of these patients are still off on sick leave on day 60). An important methodological limitation of these clinical studies is the lack of a control group/comparator that may risk having biased estimates as well. 

The persistence of different types of symptoms suffered by long COVID patients and their duration, are variable from person to person. This is very important and interesting aspect to be explored. As mentioned previously, the focus of this paper was to calculate the burden of COVID-19 using all national available data in France. We did not focus on post-acute COVID-19 survivors to characterise the impact of long COVID especially in context of “all other conditions”. This would require another method adopted to long COVID and using national available data. At time the study was conducted, some studies are ongoing to collect the relevant data.

Nevertheless, to account for this important comment, we have added a more detailed paragraph in the discussion section (implications for public health; line 442-454) describing what these studies have found in context of long-COVID. 

Third, « I also request that clinical manifestations other than long-COVID, such as myocarditis and psychiatric illness, be considered. (Reviewer’s comment) ». 

In the review by Nalbandian et al, a study by Rajpal S et al, included only 26 competitive college athletes with mild or asymptomatic SARS-CoV-2 infection, cardiac MRI revealed features diagnostic of myocarditis in 15% of participants, and previous myocardial injury in 30.8% of participants. This has a very small sample size. In table 1 of Nalbandian et al review, myocarditis was not reported as the most frequent and persisting symptom by different studies.

We fully agree that clinical manifestations other than long-COVID, such as myocarditis and psychiatric illness are very important to explore the organ-specific sequelae of COVID-19 survivors with persistent symptoms. It is not easy to take into account the myocarditis and psychiatric illness in the context of long COVID and we did not perform this analysis due to two main reasons (as mentioned above) : first, we think it would require a specific method adopted to long COVID and second, lack of relevant data collected from local data sources to explore this aspect. Moreover, long COVID is a new domain of research and it would need more longitudinal studies to draw conclusions on this point. 

Comments of first round:

Lastly, I request an extension of the literature review, comparing the current estimates with those others have found. The current paper may be the first to consider both in-hospital morbidity and long-COVID, in addition to death (many others explicitly exclude long-COVID, e.g. https://www.researchsquare.com/article/rs-1026794/v2) however it estimates a much higher share of YLD than https://www.jclinepi.com/article/S0895-4356(21)00339-5/fulltext . Please compare your findings to those of the latter paper, as well as any others you may find that compute DALY for long-COVID.

Response

Just a remark that we have cited relevant articles and compared the findings with those that applied the same disability weight and same disease model.

The latter paper (https://www.jclinepi.com/article/S0895-4356(21)00339-5/fulltext ) has applied disability weights used for chronic fatigue syndrome (CFS) for long COVID and has applied a different model than the European Burden o Disease consensus model. One of the main limitations of that study was the use of disability weights for CFS and long COVID has symptoms that CFS does not, leading to underestimation of the true disability weight of long COVID. 

Comments of second round:

Authors did not implement the additional analyses I requested: instead they added a few sentences saying that they hadn't done so. They easily could have done those analyses with existing data.

Without those additional analyses the paper substantially misrepresents the situation and has potential to mislead policy if published as it stands. It could most charitably be described as incomplete / biased. 

Please make the changes I requested in the previous round of edits, especially the sensitivity analysis using numbers for long-COVID prevalence obtained from contacting patients.

Response: We agree that we did not fully perform the additional analyses as you suggested. Nevertheless, we have integrated an additional sensitivity analysis by increasing the duration of long-COVID to 140 days in the main results that highlighted the impact of duration of long COVID on YLD estimate, which was rather easy. 

As mentioned above, we fully agree with all your comments related to the persistence of different symptoms and the clinical manifestations other than long COVID among COVID survivors. Indeed, these are very important aspects in the context of long COVID. It would require another study with a specific method adopted to characterise the impact of long COVID, using local data sources, reflecting the French context.

---

## [Decision Letter · Decision Letter 2]

15 Nov 2022

PONE-D-22-11149R2Direct impact of COVID-19 by estimating disability-adjusted life years at national level in France in 2020PLOS ONE

Dear Dr. Haneef,

Thank you for submitting your manuscript to PLOS ONE. After careful consideration, we feel that it has merit but does not fully meet PLOS ONE’s publication criteria as it currently stands. Therefore, we invite you to submit a revised version of the manuscript that addresses the points raised during the review process.

One of the reviewers still requests some minor revisions. Please address these before resubmitting. ==============================

We look forward to receiving your revised manuscript.

Kind regards,

Joël Mossong, PhD

Academic Editor

PLOS ONE

Journal Requirements:

Reviewers' comments:

Reviewer's Responses to Questions

**Comments to the Author**

1. If the authors have adequately addressed your comments raised in a previous round of review and you feel that this manuscript is now acceptable for publication, you may indicate that here to bypass the “Comments to the Author” section, enter your conflict of interest statement in the “Confidential to Editor” section, and submit your "Accept" recommendation.

Reviewer #2: (No Response)

2. Is the manuscript technically sound, and do the data support the conclusions?

Reviewer #2: Partly

3. Has the statistical analysis been performed appropriately and rigorously? 

Reviewer #2: Yes

4. Have the authors made all data underlying the findings in their manuscript fully available?

Reviewer #2: Yes

5. Is the manuscript presented in an intelligible fashion and written in standard English?

Reviewer #2: Yes

6. Review Comments to the Author

Reviewer #2: Dear Editor,

Thank you for the opportunity to review this paper. My major concern is the assumption that long-COVID duration never exceeds 140 days, an assumption which is already disproven. However, the this assumption is based on an external source (European Burden of Disease Network consensus disease model) which is likely to be widely used, and this the paper fills a needed gap in the literature.

I request that this assumption be highlighted in Abstract and perhaps also in Discussion, as follows:

Abstract:

Background The World Health Organization declared a pandemic of coronavirus disease 2019 (COVID-19), caused by severe acute respiratory syndrome coronavirus 2 (SARS-CoV2), on March 11, 2020. The standardized approach of disability-adjusted life years (DALYs) allows for quantifying the combined impact of morbidity and mortality of diseases and injuries. The main objective of this study was to estimate the direct impact of COVID-19 in France in 2020, using DALYs to combine the population health impact of infection fatalities, acute symptomatic infections and their post-acute consequences in the 140 days following the initial infection.

Methods National mortality, COVID-19 screening, and hospital admission data were used to calculate DALYs based on the European Burden of Disease Network consensus disease model. Scenario analyses were performed by varying the number of symptomatic cases and duration of symptoms up to a maximum of 140 days, defining COVID-19 deaths using the underlying, and associated, cause of death.

Results In 2020, the estimated DALYs due to COVID-19 in France were 990 710 (1472 per 100 000), with 99% of burden due to mortality (982 531 years of life lost, YLL) and 1% due to morbidity (8179 years lived with disability, YLD) in the 140 days following infection. The contribution of YLD due to acute symptomatic infections among people younger than 70 years was higher (67%) than among people aged 70 years and above (33%). Post-acute consequences contributed to 49% of the total morbidity burden. YLL among people aged 70 years and above, contributed to 74% of the total YLL. Conclusions COVID-19 had a substantial impact on population health in France in 2020. The majority of population health loss was due to mortality. Men had higher population health loss due to COVID-19 than women. Post-acute consequences of COVID-19 had a large contribution to the YLD component of the disease burden, even in the scenario where long-COVID duration is limited to 140 days. Further research is recommended to assess the impact of health inequalities associated with these estimates.

Discussion, first paragraph:

Our study is the first to estimate DALYs associated with the direct health impact of COVID 19 in France in 2020, the first year of the pandemic. When long-COVID duration was capped at a maximum of 140 days, the majority of population health loss was due to mortality, which contributed to 99% to the estimated DALY. People aged 70 years and 16 above had higher health loss due to mortality when compared to people aged younger than 70 years. On the contrary, people aged younger than 70 years had higher disability due to acute COVID-19 infections than those aged 70 years and above. Our analysis highlighted that even if long-COVID resolves after less than six months, the post-acute consequences of COVID-19 had a large contribution to the YLD component of disease burden. Moreover, we observed that women had higher YLD due to the post-acute consequences of COVID-19 than men (2147 vs 1824). Other studies have also highlighted that females were more likely to have post-COVID syndrome than males [26-28]. However, population health loss due to mortality and morbidity due to COVID-19 infections, was higher among men than women.

Discussion, line 400

Fifth, the maximum duration of long-COVID is currently unknowable but is known to be at least several years. Its duration is a major determinant of YLD of post-acute consequences of COVID-19. Although our source assumes a maximum duration of 28 days, we used using a fivefold increase in duration of post-acute consequences (i.e., 140 days) as a scenario analysis, to evaluate the impact of a much longer duration of post- acute consequences of COVID on YLD estimates. Nevertheless, the duration of post-acute consequences of COVID-19 remains uncertain and more research is needed to draw conclusions on its duration.

7. PLOS authors have the option to publish the peer review history of their article (what does this mean?). If published, this will include your full peer review and any attached files.

Reviewer #2: No

---

## [Author Response · Author response to Decision Letter 2]

1 Dec 2022

Response to reviewers’ comments

Reviewer #2: Thank you for the opportunity to review this paper. My major concern is the assumption that long-COVID duration never exceeds 140 days, an assumption which is already disproven. However, this assumption is based on an external source (European Burden of Disease Network consensus disease model) which is likely to be widely used, and this the paper fills a needed gap in the literature.

I request that this assumption be highlighted (in red) in Abstract and perhaps also in Discussion, as follows:

Response: Thank you very much for these suggestions and your kind support. 

As discussed in the previous round of revisions and mentioned in the discussion section, the duration of long-COVID is uncertain due to lack of precise data that could help to conclude the real duration of long-COVID. Therefore, to take into account the uncertainty around this duration (already mentioned in the method section-line 237-240); we applied two sensitivity analyses for duration of long-COVID of 84 days and 140 days. Over time, when more evidence will be available, then we can hope to approach a more accurate duration of long-COVID. Nevertheless, these sensitivity analyses help to evaluate the impact of the duration on YLD estimates.

Text in red: suggestions by the reviewer

Text in black, bold, italic font: Authors’ response

We have highlighted the assumptions in the abstract as well as in the discussions sections as follows suggested by the reviewer:

Abstract:

1. Background: The World Health Organization […]. The main objective of this study was to estimate the direct impact of COVID-19 in France in 2020, using DALYs to combine the population health impact of infection fatalities, acute symptomatic infections and their post-acute consequences in the 140 days following the initial infection.

Response: We have updated the text as follows because our main results of YLD (i.e., 1% contribution of YLD: 8179) included both types of infections: acute infections (i.e., mild, moderate and severe) and long-COVID (i.e., with baseline duration of 28 days up to 140 days), following the initial infection. 

« Background: The World Health Organization […]. The main objective of this study was to estimate the direct impact of COVID-19 in France in 2020, using DALYs to combine the population health impact of infection fatalities, acute symptomatic infections and their post-acute consequences, in 28 days (baseline) up to 140 days, following the initial infection. »

2. Methods: National mortality, COVID-19 screening, and hospital admission data were used to calculate DALYs based on the European Burden of Disease Network consensus disease model. Scenario analyses were performed by varying the number of symptomatic cases and duration of symptoms up to a maximum of 140 days, defining COVID-19 deaths using the underlying, and associated, cause of death. 

Response: The text is updated as suggested. 

« Methods: [….]. Scenario analyses were performed by varying the number of symptomatic cases and duration of symptoms up to a maximum of 140 days, defining COVID-19 deaths using the underlying, and associated, cause of death. » 

3. Results: In 2020, the estimated DALYs due to COVID-19 in France were 990 710 (1472 per 100 000), with 99% of burden due to mortality (982 531 years of life lost, YLL) and 1% due to morbidity (8179 years lived with disability, YLD) in the 140 days following infection. 

Response: As mentioned earlier, our baseline duration of long-COVID was 28 days according to our disease model. Our main results of YLD (i.e., 1% contribution of YLD: 8179) included both types of infections: acute infections (i.e., mild, moderate and severe) and long-COVID (i.e., with baseline duration of 28 days). The duration of 140 days for long-COVID is part of one of the sensitivity analyses. We updated the text as follows: 

« In 2020, the estimated DALYs due to COVID-19 in France were 990 710 (1472 per 100 000), with 99% of burden due to mortality (982 531 years of life lost, YLL) and 1% due to morbidity (8179 years lived with disability, YLD), following the initial infection. The contribution of YLD reached 375%, assuming the duration of 140 days of post-acute consequences of COVID-19. »

4. Conclusions: COVID-19 had a substantial impact on population health in France in 2020. The majority of population health loss was due to mortality. Men had higher population health loss due to COVID-19 than women. Post-acute consequences of COVID-19 had a large contribution to the YLD component of the disease burden, even in the scenario where long-COVID duration is limited to 140 days. Further research is recommended to assess the impact of health inequalities associated with these estimates.

Response: As mentioned earlier, our baseline duration of long-COVID is 28 days, even with this shortest duration, the long-COVID burden is large and with 140 days (part of sensitivity analysis), long-COVID burden was increased by 375%. The text is updated accordingly. 

« Conclusions: [….]. Post-acute consequences of COVID-19 had a large contribution to the YLD component of the disease burden, even when we assume the shortest duration of 28 days, long COVID burden is large. Further research is recommended to assess the impact of health inequalities associated with these estimates. »

Discussion, first paragraph:

5. Our study is the first to estimate DALYs associated with the direct health impact of COVID 19 in France in 2020, the first year of the pandemic. When long-COVID duration was capped at a maximum of 140 days, the majority of population health loss was due to mortality, which contributed to 99% to the estimated DALY. 

Response: As mentioned above, the main results of YLD (8179) are calculated based on acute infections (i.e., mild, moderated and severe) and long-COVID (with a baseline duration of 28 days). We updated the text as follows (line 334):

“The majority of population health loss was due to mortality, which contributed to 99% to the estimated DALY. This finding is in the context of long COVID being capped at a maximum of 140 days, but would still hold if long COVID duration was uncapped for the entire year.”

6. Our analysis highlighted that even if long-COVID resolves after less than six months, the post-acute consequences of COVID-19 had a large contribution to the YLD component of disease burden. 

Response: We have updated the text as suggested (line 340). 

Our analysis highlighted that even if long COVID resolves after less than six months, the post-acute consequences of COVID-19 had a large contribution to the YLD component of disease burden. 

Discussion, line 400

7. Fifth, the maximum duration of long-COVID is currently unknowable but is known to be at least several years. Its duration is a major determinant of YLD of post-acute consequences of COVID-19. Although our source assumes a maximum duration of 28 days, we used a fivefold increase in duration of post-acute consequences (i.e., 140 days) as a scenario analysis, to evaluate the impact of a much longer duration of post- acute consequences of COVID on YLD estimates. 

Response: Thanks for these inputs. We have updated the text as follow (line 406):

Fifth, the maximum duration of long COVID is currently unknown but a subset of patients could suffer these consequences through the entire annual period of study and we still lack the data to assess these consequences. The duration of long COVID is a major determinant of YLD of post-acute consequences of COVID-19. Our disease model was based on a duration of 28 days, we used a threefold and fivefold (i.e., 84 days and 140 days) increase in duration of post-acute consequences, as sensitivity analysis, to evaluate the impact of a much longer duration of post- acute consequences of COVID on YLD estimates.

---

## [Decision Letter · Decision Letter 3]

13 Jan 2023

Direct impact of COVID-19 by estimating disability-adjusted life years at national level in France in 2020

PONE-D-22-11149R3

Dear Dr. Haneef,

We’re pleased to inform you that your manuscript has been judged scientifically suitable for publication and will be formally accepted for publication once it meets all outstanding technical requirements.

Kind regards,

Joel Mossong, PhD

Academic Editor

PLOS ONE

Additional Editor Comments (optional):

Reviewers' comments:

Reviewer's Responses to Questions

**Comments to the Author**

1. If the authors have adequately addressed your comments raised in a previous round of review and you feel that this manuscript is now acceptable for publication, you may indicate that here to bypass the “Comments to the Author” section, enter your conflict of interest statement in the “Confidential to Editor” section, and submit your "Accept" recommendation.

Reviewer #2: All comments have been addressed

2. Is the manuscript technically sound, and do the data support the conclusions?

Reviewer #2: Yes

3. Has the statistical analysis been performed appropriately and rigorously? 

Reviewer #2: Yes

4. Have the authors made all data underlying the findings in their manuscript fully available?

Reviewer #2: Yes

5. Is the manuscript presented in an intelligible fashion and written in standard English?

Reviewer #2: Yes

6. Review Comments to the Author

Reviewer #2: (No Response)

7. PLOS authors have the option to publish the peer review history of their article (what does this mean?). If published, this will include your full peer review and any attached files.

Reviewer #2: **Yes: **Maia P. Smith

---

## [Editor Report · Acceptance letter]

16 Jan 2023

PONE-D-22-11149R3 

Direct impact of COVID-19 by estimating disability-adjusted life years at national level in France in 2020 

Dear Dr. Haneef:

I'm pleased to inform you that your manuscript has been deemed suitable for publication in PLOS ONE. Congratulations! Your manuscript is now with our production department. 

Kind regards, 

on behalf of

Dr. Joel Mossong 

Academic Editor

PLOS ONE